# Stimulus duration encoding occurs early in the moth olfactory pathway
Tomas Barta [1,2,3] ✉, Christelle Monsempès[1], Elodie Demondion[1], Abhishek Chatterjee[1], Lubomir Kostal [2,4] ✉ & Philippe Lucas [1,4] ✉

Pheromones convey rich ethological information and guide insects' search behavior. Insects navigating in turbulent environments are tasked with the challenge of coding the temporal structure of an odor plume, obliging recognition of the onset and offset of whiffs of odor. The coding mechanisms that shape odor offset recognition remain elusive. We designed a device to deliver sharp pheromone pulses and simultaneously measured the response dynamics from pheromone-tuned olfactory receptor neurons (ORNs) in male moths and *Drosophila*. We show that concentration-invariant stimulus duration encoding is implemented in moth ORNs by spike frequency adaptation at two time scales. A linear-nonlinear model fully captures the underlying neural computations and offers an insight into their biophysical mechanisms. *Drosophila* use pheromone *cis*-vaccenyl acetate (cVA) only for very short distance communication and are not faced with the need to encode the statistics of the cVA plume. Their cVA-sensitive ORNs are indeed unable to encode odor-off events. Expression of moth pheromone receptors in *Drosophila* cVA-sensitive ORNs indicates that stimulus-offset coding is receptor independent. In moth ORNs, stimulus-offset coding breaks down for short (< 200 ms) whiffs. This physiological constraint matches the behavioral latency of switching from the upwind surge to crosswind cast flight upon losing contact with the pheromone.

Flying insects rely heavily on olfactory cues to search for potential mates, food, and oviposition sites. However, turbulent airflow breaks the odor signal (e.g., sex pheromone from a female) into pockets containing odor and pockets with clean air. Therefore the insect can encounter pockets with a high concentration of pheromone even at large distances from the female[1–4]. The odor plume does not form a continuous gradient pointing to its source, and obtaining a reliable concentration average would take too long for flying insects to efficiently track odor plumes. Instead, the insect must implement different searching strategies, such as surging upwind during an odor encounter and crosswind casting when the odor signal is lost[5–9]. This searching strategy requires the ability to reliably detect the onset and offset of the odor pocket.

Some olfactory receptor neurons (ORNs) respond to odor stimulus with a biphasic response pattern, thus clearly marking the onset of the stimulus with a transient peak firing activity and the offset with transient inhibition[10–13]. However, the odor offset-marking transient inhibition has so far been observed only with odor molecules that have high volatility. A

theoretical model predicts that many ORNs in *Drosophila* encode odor with the same dynamics and the differences in response patterns can be almost fully explained by the differences in odor-delivery precision. However, this prediction has not yet been verified experimentally, and moreover, cannot hold universally, as the same odor molecules can lead to both phasi-tonic and tonic response patterns, depending on the odor receptor (OR) present in the stimulated ORN[11].

A striking example of ORNs where the biphasic pattern has not been observed, namely termination of the response with stimulus offset, are the pheromone sensitive ORNs in male moths[14–17], a classical model for odor-guided navigation in turbulent environments due to their ability to track the pheromone plumes at large distances[18–21]. The apparent inability of moth ORNs to detect the pheromone stimulus offset is very surprising, given the rich and complex repertoire of maneuvers they exhibit when navigating pheromone plumes[9,22,23].

Compared to plant volatile compounds, pheromones have relatively low volatility, as indicated by their low vapor pressure[24], and when used as

[1]Department of Sensory Ecology, Institute of Ecology and Environmental Sciences of Paris, INRAE, Sorbonne Université, CNRS, IRD, UPEC, Université de Paris, Route de Saint Cyr, Versailles, 78000, France. [2]Laboratory of Computational Neuroscience, Institute of Physiology of the Czech Academy of Sciences, Vídeňská 1083, Prague, 14220, Czech Republic. [3]Neural Coding and Brain Computing Unit, Okinawa Institute of Science and Technology, 1919-1 Tancha, Onna, 904-0412 Okinawa, Japan. [4]These authors jointly supervised this work: Lubomir Kostal and Philippe Lucas. ✉e-mail: tomas.barta@oist.jp; kostal@biomed.cas.cz; philippe.lucas@inrae.fr

olfactory stimuli they are likely to exhibit slower dynamics. Therefore, we investigated whether the slow response termination is a physiological property of ORNs, or an artifact caused by interactions of pheromone molecules with the odor-delivery device.

Similarly, to gain a further insight into the dynamics of ORN responses to complex pheromone molecules, we investigated the response dynamics of *Drosophila* ORNs in the T1 sensilla to the pheromone *cis*-vaccenyl acetate (cVA). Unlike moth pheromone sensitive neurons, and widely studied *Drosophila* volatile plant compound (VPC) sensitive ORNs, cVA sensitive ORNs are not used for guiding mid to long-range navigation, but only for identification on very short distances[25,26]. Being able to compare the responses of ORNs with different behavioral significance helps us understand why the response patterns evolved a certain way.

Using a new odor-delivery system, we observed a tri-phasic pattern in the ORN responses from the moth species *Agrotis ipsilon* and *Spodoptera littoralis*. This pattern consisted of an excitatory response at stimulus onset, an inhibitory phase at stimulus offset, and a less intense excitatory activity (rebound activity) following the inhibitory phase. This contrasted the widely held belief that responses to pheromone in moth ORNs terminate very slowly and was reminiscent of the response profile of projection neurons (PNs). Yet, when ORNs were subjected to short stimuli, the inhibitory phase disappeared, and the response consisted of a single long-lasting burst that significantly exceeded the stimulus duration. Therefore, the moth ORNs are capable of encoding the stimulus duration, but only for sufficiently long stimuli.

The observed qualitative differences in the response (i.e., mono-phasic response to short stimuli and tri-phasic response to long stimuli) point to slow adaptation of the ORNs. To assess the slow adaptation process, we isolated the ORN processing capabilities from the dynamics of the odor delivery. We measured the local field potential (LFP) in the sensilla, which is tightly correlated with the depolarizing current entering the ORN. We recorded both the LFP and the firing response to study independently the transduction processes leading to the generation of the receptor current and how the spike-generating mechanism in the soma responds to this current[11]. We performed an optimization procedure to narrow down the adaptation processes to only two time scales, which provided novel insights into the mechanisms responsible for the adaptation and the firing response shape.

The *Drosophila* ORNs responded tonically to the cVA stimulus, without an inhibitory phase marking the stimulus offset. We observed the same tonic response dynamics in mutant flies expressing the moth pheromone sensitive OR. This suggests that the spike generator of the T1 sensilla ORNs acts significantly differently to the spike generator in the *Drosophila* ORNs used for long range navigation and moth ORNs. Since cVA is not used for long range communication, this hints that the spike generator detecting odor onset and offset in moth and *Drosophila* ORNs evolved to support long range communication and navigation.

## Results
### New odor-delivery device improves the speed of odor onset and offset
A common type of odor-delivery device in insect olfactory studies consists of Pasteur pipettes containing a filter paper loaded with one of the odors/doses to test. An electrovalve (EV) redirects an airstream through the pipette, the small tip of which is introduced into a hole on the side of a glass tube that bathes the insect antenna with a constant humidified and filtered airstream[27–29]. However, the time constants of rising and falling odor concentrations at the onset and offset of the stimulus can be very long, depending on the physicochemical properties of the odorant[12,30,31]. First, odors are sticky, and adsorption/desorption on surfaces contributes to low-pass filtering of the stimulus dynamics as the odors travel along the tube. Second, the temporal structure of the odor stimuli disintegrates within 10–20 mm from the exit of the odor stimulus device when the airflow is no more restrained within the tubing.

Therefore, we built an odor-delivery device in which we reduced the surface that can adsorb odor molecules to minimize their effect on the dynamics of the delivered stimulus. The insect was placed directly in front of

an electrovalve controlling the odorant supply. To test whether the odor-delivery device is capable of delivering sharp and short odor pulses, we used linalool as a proxy for pheromone because linalool has a relatively low volatility but can be monitored with a photo-ionization detector (PID) (Fig. 1A). The addition of a glass tube between the PID and the electrovalve (15 cm length, 5 mm internal diameter) resulted in much slower PID responses, and short stimuli evoked very little response (Fig. 1B).

More volatile compounds (acetone, α-pinene) triggered sharper PID responses (Fig. 1C). We suspected that the slowdown of the response dynamics with linalool was not a property of the odor-delivery device but of the PID. To verify this, we performed an experiment where we completely and suddenly cut off the odor-delivery device from the PID by dropping a plastic barrier between them during the stimulation. The time course of the PID response offset remained slow (Fig. 1D, E). Although the observed PID response offset was slightly faster in the first 500 ms after stimulus termination in the experiment using the plastic barrier, after 500 ms the sustained response was identical (Fig. 1E–J), indicating that the observed slow dynamics of the response and the long-lasting response were mostly a property of the PID and not of the odor-delivery device. Possibly the odorant molecules adhere to the surface of the PID and thus slow down their onset and offset detection by the PID. Therefore, we conclude that it is risky to use a PID signal as a proxy for odor stimulus dynamics, and the physiochemical properties of the used odorant need to be considered.

### Moth ORN response shape tracks odor pulse durations
We presented the pheromone-sensitive ORNs of *A. ipsilon* with stimuli of different durations (3, 5, 10, 20, 50, 100, 200, 500 ms, 1 s, 2 s, and 5 s) of the 100 pg dose. The neurons responded phasi-tonically: the ORNs reached the peak of their firing activity within 50 ms after the stimulus onset, and then the firing activity started decreasing towards a steady-state level. The time course of the response changed qualitatively with the stimulus duration (Fig. 2A, B). For a stimulus duration below 100 ms the neurons continued firing for around 100 ms after the stimulus offset, while slowly returning to their spontaneous activity (Fig. 2C, D). This can be seen as a pulse response with a stereotypical shape, which changes very little with the stimulus duration and exceeds the stimulus duration (Supplementary Fig. 1). For stimuli longer than 200 ms the firing response terminated sharply with the stimulus offset. The firing response was then followed by an inhibitory phase. To capture the inhibitory phase, we calculated the number of action potentials in the interval 100–400 ms after the firing response termination, where the inhibitory phase appeared the most pronounced for long stimuli. We compared the rate during the inhibitory phase with the rate during the rebound activity, which we calculated in the interval 1–3 s, where the ORNs appeared to have fully recovered from the inhibitory phase. As expected, we observed a significant difference in the firing rates for long stimuli (two-tailed Wilcoxon rank test, 200 ms: $T = 46$, $n = 22$, $p \doteq 0.027$; 500 ms: $T = 46$, $n = 23$, $p \doteq 0.004$; 1 s: $T = 13$, $n = 22$, $p < 0.001$; 2 s: $T = 0$, $n = 22$, $p < 0.001$; 5 s: $T = 1$, $n = 22$, $p < 0.001$), while we found no significant difference in the firing rates for short stimuli (two-tailed Wilcoxon rank test, 3 ms: $T = 10$, $n = 7$, $p \doteq 0.92$; 5 ms: $T = 15$, $n = 13$, $p \doteq 0.67$; 10 ms: $T = 62$, $n = 21$, $p \doteq 0.76$; 20 ms: $T = 42$, $n = 20$, $p \doteq 0.31$; 50 ms: $T = 82.5$, $n = 20$, $p \doteq 0.90$; 100 ms: $T = 58$, $n = 22$, $p \doteq 0.14$). The rebound activity increased with stimulus duration, making the inhibitory phase more pronounced (Fig. 2F). We noted heterogeneity in the firing responses across ORNs, as also reported previously[17]. However, all response patterns had the same general shape (Supplementary Fig. 2).

A mono-phasic response to short stimuli and inhibitory phase after long stimuli were also observed with higher (1 ng) and lower (10 pg) pheromone doses (Fig. 3A, B; Wilcoxon rank test for differences between firing rates during inhibitory phase and rebound phase—10 pg and 20 ms: $T = 61$, $n = 28$, $p \doteq 0.10$; 200 ms: $T = 38$, $n = 32$, $p \doteq 0.007$; 2 s: $T = 55.5$, $n = 32$, $p < 0.001$; 100 pg and 20 ms: $T = 85.5$, $n = 33$, $p \doteq 0.013$; 200 ms: $T = 235$, $n = 44$, $p \doteq 0.030$; 2 s: $T = 11$, $n = 38$, $p < 0.001$; 1 ng and 20 ms: $T = 321.5$, $n = 40$, $p \doteq 0.86$; 200 ms: $T = 47$, $n = 37$, $p < 0.001$; 2 s: $T = 36$, $n = 41$, $p < 0.001$). The contrast between the rebound activity and the inhibitory activity grew with the

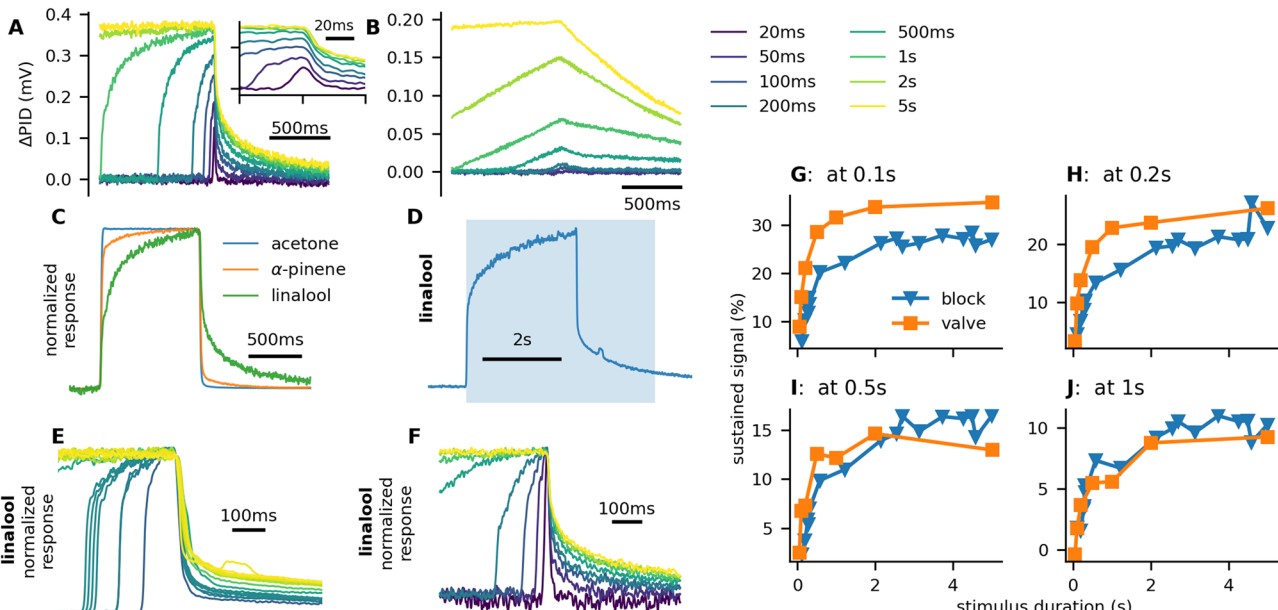

**Fig. 1 | Dynamics of the new odor-delivery device. A** We verified using the PID response to linalool that the odor-delivery device was capable of delivering sharp and short odor pulses. **B** On the contrary, adding a 15 cm glass tube after the valve produced responses that were much less sharp, and short stimuli (up to 200 ms) evoked very little PID response or no response at all (we used pure linalool instead of 10% dilution to compensate for airflow mixing in the glass tube). **C** More volatile compounds produced sharper PID responses. **D** Shaded area indicates linalool stimulation. Approximately 2.8 s after the stimulus onset a plastic barrier was dropped between the PID and the odor-delivery device to prevent any odor molecules from the odor-delivery device from reaching the PID. The offset of the PID signal remained slow. **E** We dropped the barrier at different times after the stimulus onset.

The longer the stimulus was, the slower the PID response offset. We observed the same pattern when we used our odor-delivery device to deliver stimuli of different durations (**F**). **G–J** We compared the value (averaged in a 20 ms window) of the PID at different times after the stimulus offset to its peak value. At 0.5 s after the stimulus termination, the sustained signal was the same regardless of whether the stimulus was terminated with the electrovalve or mid-odor delivery with a plastic barrier. This shows that most of the slow dynamics observed with the PID were due to the properties of the PID and not the odor-delivery device. The linalool concentration delivered was, therefore, likely to be sharper than measured by the PID. All PID responses in the figure were filtered with 49 Hz 2-pole Butterworth lowpass filter to remove noise.

stimulus dose (Fig. 3B; Spearman correlation coefficient, 20 ms: $r \doteq 0.22$, $p \doteq 0.027$; 200 ms: $r \doteq 0.37$, $p < 10^{-3}$; 2 s: $r \doteq 0.50$, $p < 10^{-5}$). Moreover, in the dose range 10 pg to 1 ng, the shape of the firing profile was mostly independent of pheromone concentration (Fig. 3C), a property that has been illustrated on *Drosophila* ORNs but only with highly volatile odors and may aid intensity invariant odor identity coding[12].

Flying insects use both olfactory and mechanosensory input (from wind speed) to track odor plumes. AL neurons integrate both of these sensory inputs[32,33]. The detection of mechanosensory information in insect antennae is attributed primarily to Johnston's organ and Böhm's bristles in the pedicel of the antenna[34–36]. However, it was recently proposed in the honeybee that mechanosensory signals can also be transduced by olfactory sensilla on the antenna, with changes of sensilla position potentially modulating the ORN responses[37]. To verify that the observed response pattern is not an artifact caused by a change in mechanical pressure at the stimulus offset, we performed recordings where we maintained a constant mechanical pressure throughout odor stimuli by delivering odorless air with an electrovalve in opposing phase to the valve controlling the odor delivery. With this setting, we still observed the tri-phasic response pattern (Supplementary Fig. 3).

We also saw the same response patterns with the ORNs of *S. littoralis* (Fig. 4). These results lead us to conclude that the previously reported sustained pheromone responses of the moth ORNs are an artifact caused by interactions of the odor molecules with the tubing of the odor-delivery device and should not occur in nature when the moth is flying sufficiently far away from any surfaces that could release previously adsorbed pheromone molecules.

We still observed some sustained activity long after the stimulus end, with onset after the inhibitory phase. The intensity of the activity increased both with the duration and dose of the stimulus (Fig. 3B) and could last more than 15 min (Supplementary Fig. 4). Our new setup strongly reduces the surfaces where odor molecules can adsorb and then desorb and stimulate the antenna, therefore, the sustained response likely has a physiological

origin (e.g., pheromone molecules adhering to the antenna before eventually reaching the odor receptors or slower signaling pathway).

We stimulated the ORNs located in the T1 sensilla of *Drosophila*. In wild-type flies, these ORNs are sensitive to the sex pheromone cVA, but not to Z7-12:Ac (Supplementary Fig. 5). We stimulated the wild-type flies with cVA, and the mutant flies expressing AipsOR3 with Z7-12:Ac. In both cases, the ORNs responded tonically, without any peak at the stimulus onset and without transient inhibition after the stimulus offset (Fig. 5A). This illustrates that the tonicity of response to cVA was not due to imprecise odor delivery and that the phasi-tonic response shape was not a property of the odor receptors, but rather a property of the spike generating mechanism, as illustrated previously in moths[38,39] and *Drosophila*[11,40]. Moreover, we also observed a phasi-tonic response of moth ORNs when stimulated with the VPC (Z)-3-hexenyl acetate (Fig. 5B), to which the pheromone-sensitive moth ORNs respond, but the response is likely mediated by a different receptor[41]. This experiment further indicated that the source of the phasi-tonicity lay in the spike-generating mechanism of the moth ORNs

### Rapid response termination stems from slow spike frequency adaptation

We recorded the LFP simultaneously with the firing activity in response to 20 ms, 200 ms, and 2 s stimuli (dose 1 ng). The LFP shape reflects the depolarizing current flowing from the sensillar lymph into the neuron (with a multi-compartment model of the ORN we estimated that the LFP corresponds to the depolarizing current filtered with an exponential kernel with 10 ms decay, Fig. 6A). After the stimulus onset, the LFP decreased (downward deflection of the LFP signal) due to the positive charge flowing from the sensillar lymph into the ORN (exciting the neuron). The LFP typically exhibited some level of adaptation (upward deflection) followed by an additional downward deflection (became more negative; Fig. 6B–E). Shortly after the stimulus offset (within 10 ms), the LFP exhibited an upward

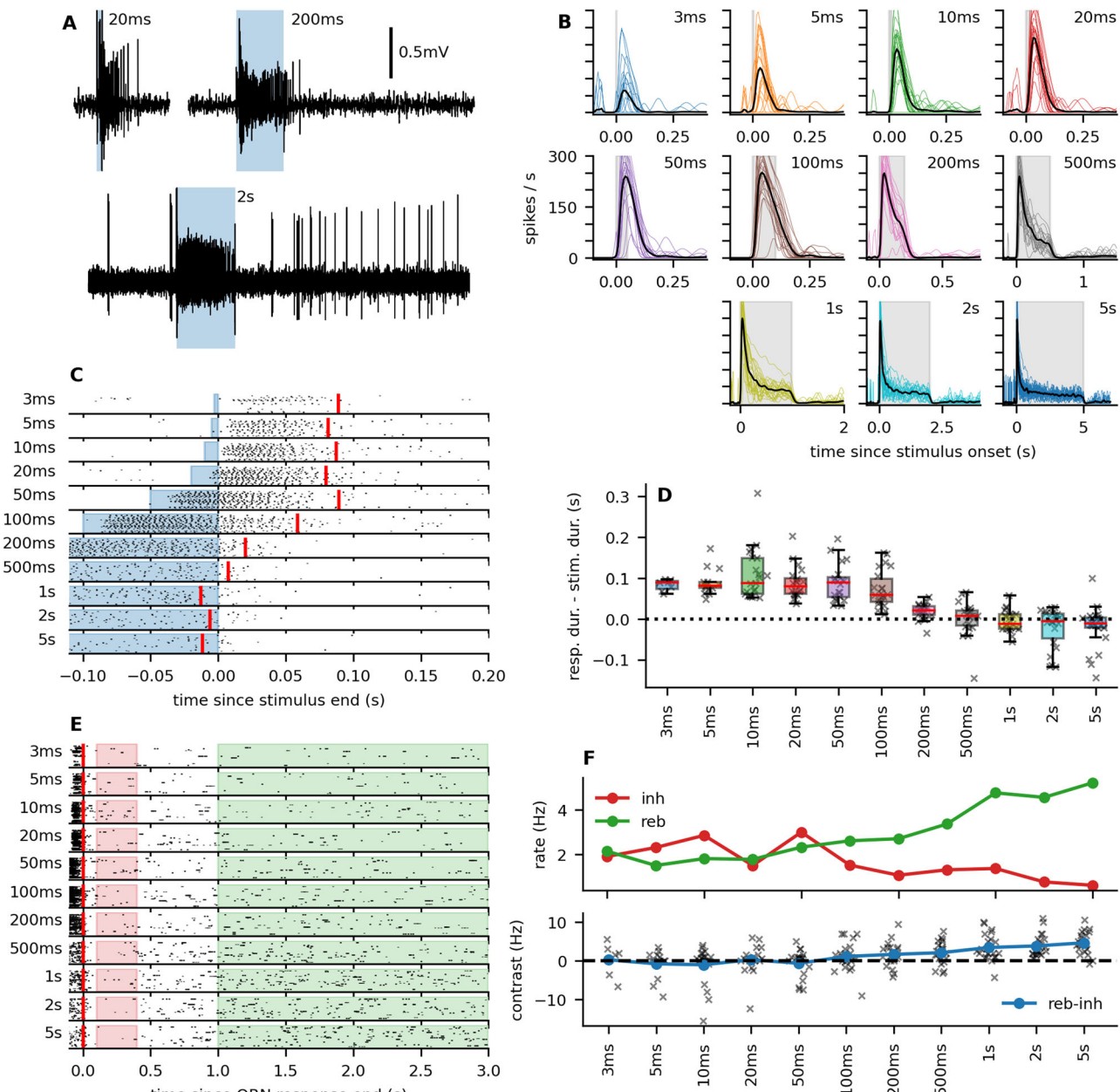

**Fig. 2 | Different stimulus durations produce qualitatively different response terminations. A** Representative voltage traces in response to 20 ms, 200 ms, and 2 s stimuli. **B** Firing responses of the ORNs to stimuli of different durations. Colored lines represent the responses of individual neurons. The black line is the average response across all recorded neurons (the shaded area indicates the stimulus period, $N = 21–23$ sensilla). **C** Raster plots of the spike trains, aligned at the stimulus offset. Responses to stimuli of 100 ms and shorter continue after the stimulus offset, while the ends of responses to longer stimuli coincide with the stimulus offset. The red vertical line represents the point in time when 50% of the ORNs' responses finished

(see "Materials and methods"). **D** Box-plot of how much the response ends exceeded the stimulus duration. The stimulus duration is color-coded, the same as in (**B**). **E** Raster plots aligned to the median response end. We compared the firing rates in the red-filled area (0.1–0.4 s after the response end) to the firing rates in the green-filled area (1–3 s after the response end) to evaluate the contrast between the inhibitory phase and the rebound activity, as shown in (**F**) (top panel: firing rate during inhibitory/rebound phase, bottom panel: the difference between the rebound and inhibitory activity; stars indicate Wilcoxon rank test significance levels *$p < 0.05$, **$p < 0.01$, ***$p < 0.001$, see the main text for test statistics and $p$-values).

deflection (became less negative), signifying a decrease in the depolarizing current. After an initial rapid upward deflection, the LFP returned very slowly toward the level before the stimulus, likely leading to the sustained firing activity. Similarly as in the case of the firing responses, we observed heterogeneity in the recorded LFP across ORNs. However, the general shape, as described above, was ubiquitous (Supplementary Fig. 2).

The transiency of the firing response indicated that the firing rate depended on the slope of the depolarizing current, as previously observed in *Drosophila*[11]. However, dependency purely on the LFP and its slope cannot fully explain the shape of the firing rate. Particularly, the average LFP response

to 200 ms and 2 s was nearly identical in the period of 50 ms before stimulus termination and to 100 ms after stimulus termination, but the lower firing rate indicated that the spike-generating mechanism was clearly more adapted after 2 s stimulation (Fig. 6F). A comparison of LFP to firing rate transformation between the response to 20 ms and the longer stimuli is not straightforward due to the weaker LFP response evoked by the 20 ms stimulus. To facilitate the comparison, we shifted the responses by 50 ms, so that the LFP decay after 20 ms stimulation closely followed the LFP decay after 200 ms stimulation, while the firing rate was significantly higher (Fig. 7A). These results illustrated a clear dependence of the firing activity on the ORN's history.

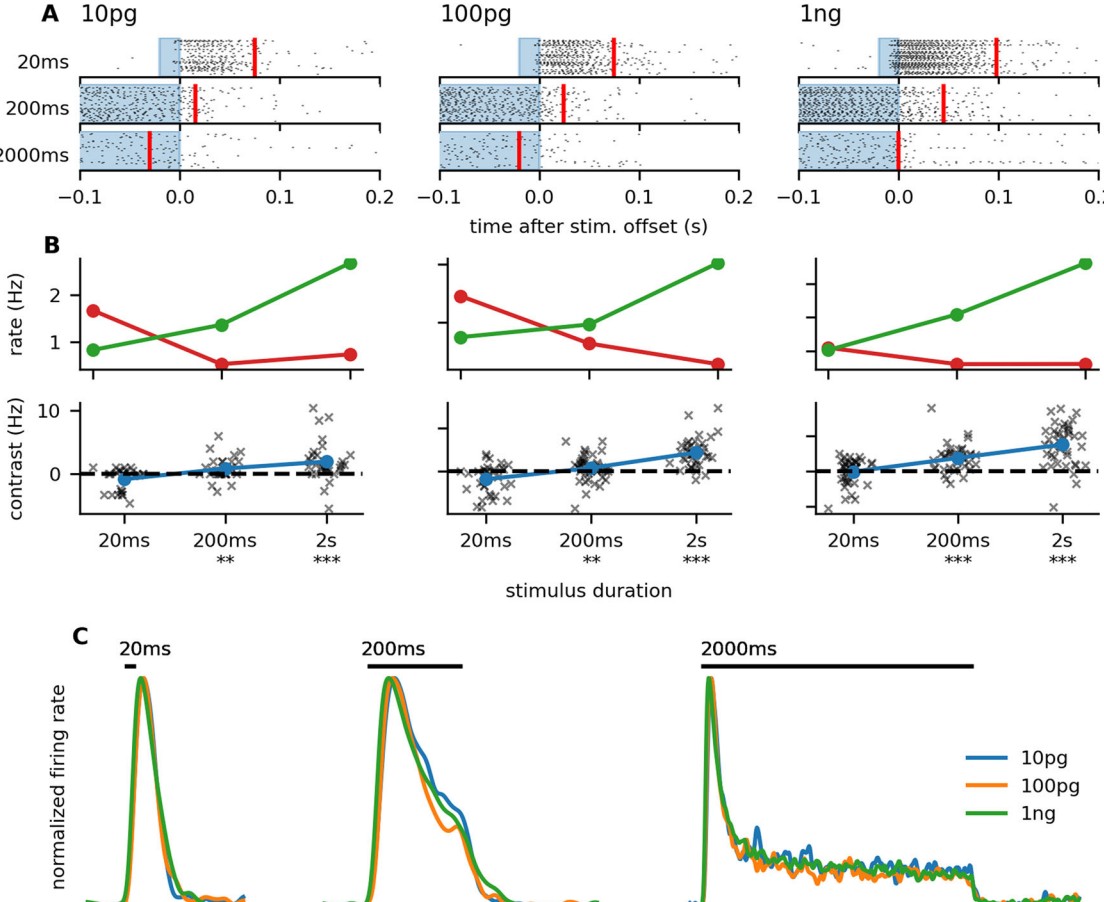

**Fig. 3 | Response properties are maintained with different odor doses. A** Raster plots aligned to the stimulus termination, as in Fig. 2C, but with different odorant doses ($N = 52$–57 sensilla). For all doses, the spiking response exceeded the short (20 ms) stimulus but terminated rapidly with the longer stimulus (2 s). The red vertical line represents the point in time when 50% of the ORNs' responses finished. **B** The equivalent of Fig. 2F for different odorant doses. With all tested doses, the neurons exhibited a transient inhibition after the 200 ms and 2 s stimuli. **C** Firing rate shapes normalized to the peak for different stimulus durations and doses. The general shape was independent of the odorant dose. The black bar indicates the stimulus presence.

Knowledge of the time scales on which ORNs integrate the input signal is essential for understanding which processes shape the firing response. To this end, we used a linear-nonlinear model to predict the firing rate from the LFP (Fig. 7A):

$$f(t) = N((K_f * \text{LFP})(t)). \tag{1}$$

The linear kernel $K_f$ is composed of multiple exponential kernels and a $\delta$-function, therefore, the convolution can be equivalently expressed as

$$K_f * \text{LFP}(t) = c_0 \cdot \text{LFP}(t) + \sum_{k=1}^{n} c_k \cdot (g_k * \text{LFP})(t) \tag{2}$$

$$g_k(t) \begin{cases} \frac{1}{\tau_k} e^{-\frac{t}{\tau_k}} & t \geq 0, \\ 0 & t < 0, \end{cases} \tag{3}$$

where $c_k$ are the linear combination coefficients, and $\tau_k$ are the time scales of the exponential kernels. $N$ is a rectifying nonlinearity ($N(x) = \max(0, x)$). The coefficients corresponding to different time scales then provide an insight into the role of the time scales in input integration. Using an optimization procedure described in "Materials and methods" (see also Supplementary Fig. 6), we found that the firing rate can be reliably predicted from the LFP using only two time scales: 31 ms and 635 ms and the unfiltered LFP (note that the LFP already provides a low-pass filtered representation of the depolarizing current).

We fitted the coefficients $c_k$ to a 2 s stimulus (and the preceding 1 s of spontaneous activity) individually to each of 26 different ORN recordings by minimizing the square error between the prediction and the observed firing rate (we fitted the model to each neuron individually, because the pheromone-sensitive ORNs of moths exhibit a significant cell-to-cell variability, as analyzed by Rospars et al.[17]). The average values of the coefficients were $c_0 = -109.2$, $c_1 = 85.8$, and $c_2 = 18.3$ (the coefficient distributions are shown in Fig. 7B). The signs indicate that the neurons respond rapidly to LFP deflection by their firing activity ($c_0 < 0$), which is then attenuated by adaptation on two different time scales ($c_k > 0$, $k \geq 1$). Using only the LFP (indicating the depolarization of the neuron) and the two adaptation time scales, we were able to predict very well the ORNs' firing responses (Fig. 7C–E). Despite being fitted only to the 2 s pulse, the predicted firing rate corresponded well also to the responses to the 20 ms and 200 ms pulses, including the firing profile after the stimulus offset, which was different for each pulse duration.

The presented model is the minimal model capable of capturing the shape of the firing response. With $c_2 = 0$ (set after the fitting procedure), the model still predicts well the response to short stimuli (during the short period, the neuron does not become adapted on the slow time scale), however, it does not predict the continued decrease of firing rate during the 2 s long stimulation.

We thus obtained a model with only three parameters, which are easily interpretable: $c_0$ is the response to depolarization, $c_1$ is the strength of the adaptation responsible for the phasicity of the response, and $c_2$ is the strength of the adaptation responsible for the gradual attenuation of the response and rapid response termination after the stimulus offset.

**Fig. 4 | Response patterns of *S. littoralis*. A** Same as Fig. 2E, but for the *S. littralis* responses. Raster plots of ORN responses to different stimulus durations, aligned to the estimated stimulus end (184, 247, and 1995 ms respectively), show that the response pattern to stimuli of different durations was the same as in *A. ipsilon*. ORNs exhibited a prolonged response to short stimuli and transient inhibition shortly after the offset of long stimuli. **B–D** Full firing profiles of responses to different stimulus durations. Horizontal black bars indicate the stimulus. **E, F** Same as Fig. 2D, F, but for the *S. littoralis* responses.

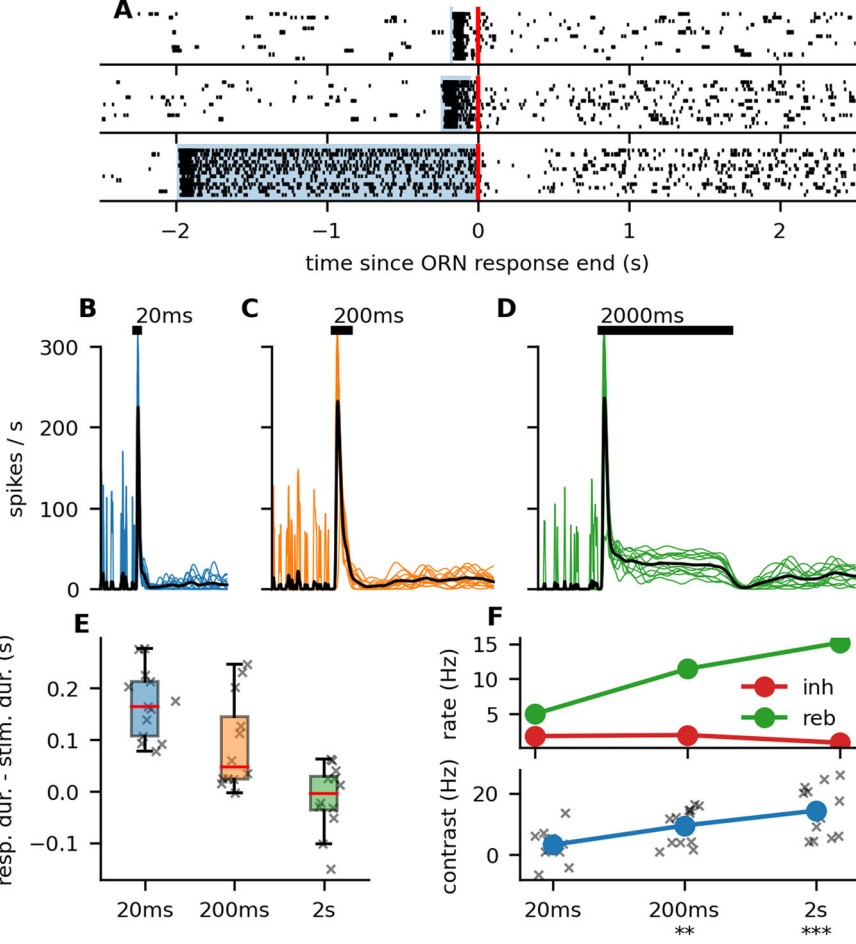

mechanism could be $Ca^{2+}$ dependent. Moth ORNs express $Ca^{2+}$-gated potassium channels (*Mamestra brassicae*[43]; *Manduca sexta*[44]; *S. littoralis*[45]). Their expression in the soma would result in hyperpolarizing currents upon their activation. We illustrated with a multicompartmental model that such hyperpolarizing currents can affect the LFP by making it more negative, despite hyperpolarizing the ORN (Fig. 8A–D) and could thus account for the second downward deflection of LFP during the 2 s stimulation. This phenomenon is the converse to what is observed with action potentials in insect ORN extracellular recordings. Action potentials appear as upward deflection of the LFP, instead of a downward deflection, because the ORN is being depolarized by current influx from the hemolymph into the soma, as opposed to current influx from the sensillar lymph into the dendrite, as is the case when receptor channels on the dendrite open. If the currents were activated by $Ca^{2+}$ entering during action potentials, the second downward deflection should be removed by abolishing the spiking activity and thus also the $Ca^{2+}$ influx due to action potentials. To test this hypothesis, we recorded the LFP after injecting the $Na^+$ channel antagonist tetrodotoxin (TTX) (50 μM) into the antenna. The TTX injections abolished the spiking activity. However, the secondary downward deflection of the LFP remained (Fig. 8E, F). Therefore, we concluded that the secondary deflection was not caused by hyperpolarizing currents in the soma triggered by $Ca^{2+}$ influx during action potentials but could still be caused by $Ca^{2+}$ influx from other sources, such as through the receptor channels.

### A minimal odor-to-firing rate model predicts firing rate shapes

It is also possible to obtain a full odor-to-firing-rate model. We used a simple transduction model to predict the LFP from the odor concentration[11]:

$$R \underset{s_b}{\overset{[O]k_b s_b}{\rightleftarrows}} OR \underset{s_a}{\overset{k_a s_a}{\rightleftarrows}} OR^*, \qquad (4)$$

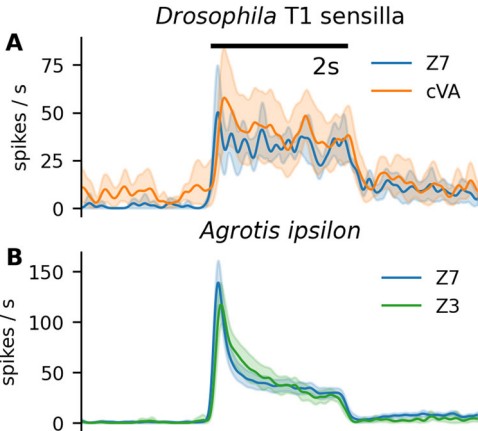

**Fig. 5 | Phasi-tonicity is not a receptor property. A** Response of ORNs in the *Drosophila* T1 sensilla in wild-type flies to 10 μg of cVA (8 ORNs) and in the mutant flies to 100 ng of Z7-12:Ac (Z7) (6 ORNs). In both cases the response is very tonic, not exhibiting any adaptation. **B** Response of the pheromone-sensitive ORNs in the *A. ipsilon* trichoid sensilla to the main pheromone compound Z7-12:Ac and to the plant volatile compound (Z)-3-hexenyl acetate (Z3) (12 ORNs). In both cases, the response is phasi-tonic, despite each molecule activating different receptors on the dendrite. The shaded areas represent 95% confidence of the mean, obtained by bootstrapping the responses. The stimuli are 2 s long.

The optimal time scales should reflect the time scales of the physiological processes responsible for the adaptation. The slow adaptation time constant of 635 ms approximately corresponds to $Ca^{2+}$ extrusion time scales (0.4–1 s in *Drosophila* ORNs[42]). This indicates that the adaptation of the spike generating

**Fig. 6 | Firing rate depends on the history of the input. A** LFP is a low-pass filtered image of the receptor current. We used a multicompartmental ORN model to simulate the measured LFP in response to a receptor current $I_E$. The shape of the LFP ($V_{ed}$, black line) coincided with the shape of the receptor current (blue line) smoothed with an exponential filter with a 10 ms time constant (dashed orange line). **B–D** Raw recordings of a single ORN's response to three different stimulus durations, recorded with a glass electrode. The blue shaded area indicates the stimulus duration (20 ms, 200 ms, 2 s from **B** to **D**). **E** LFP responses averaged over 26 sensilla. Note that in response to the 2 s stimulus, after the initial downward deflection, LFP went upwards, indicating receptor adaptation, and afterwards continued to go downward again. This was also apparent in (**D**). **F** LFP (top) and firing rate (bottom) aligned at the stimulus termination. The LFP after the stimulus offset was identical for the 200 ms and 2 s stimulus, yet their firing rates were dramatically different. The dashed blue lines indicate the response to the 20 ms stimulus but shifted by 50 ms. Then, the LFP time course after the stimulus offset was identical with the 200 ms stimulus, but again, the firing rates greatly differed.

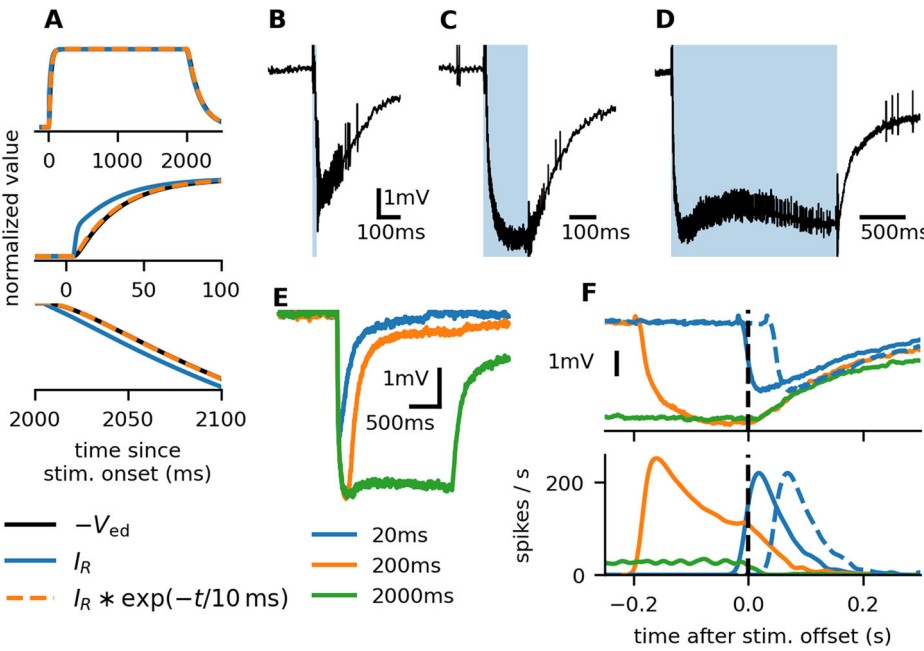

**Fig. 7 | Slow spike-frequency adaptation is necessary to reproduce the ORNs' behavior. A** Illustration of the firing rate prediction process. The LFP was filtered with two different exponential kernels with time constants $\tau_1$ and $\tau_2$. A linear combination of the filtered values and the LFP, followed by a rectifying non-linearity, provides a prediction of the firing rate. This process is equivalent to directly convoluting the LFP with a linear filter composed of two exponential kernels and a $\delta$-function. **B** Values of the optimal coefficients for all the fitted neurons. Points are color-coded by ORNs. **C–E** Predictions of the firing rate with and without the slow (800 ms) component. Predictions with the full filter closely match the empirical firing rate (dashed black line). The reduced filter predicts well the responses to short stimuli but fails to predict the response to the 2 s stimulus.

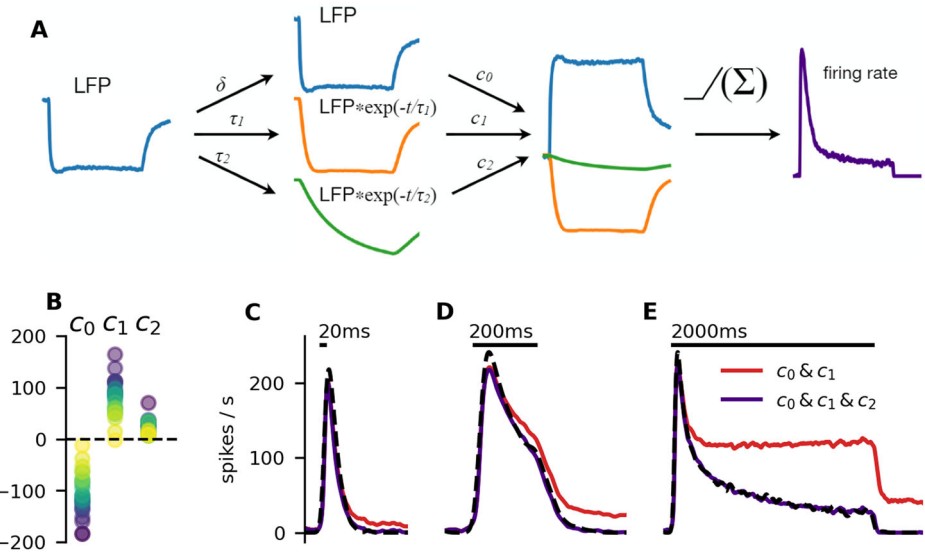

$$\text{LFP} = \text{OR}^* * g_{\text{LFP}}, \qquad (5)$$

where R are the unbound receptors, OR are bound but not activated receptors, OR* are bound and activated receptors, [O] is the odorant concentration, $s_a$ and $s_b$ are the unbinding and deactivation rates, and $k_a$ and $k_b$ set the ratio between activation/deactivation and binding/unbinding rates and $g_{\text{LFP}}$ is an exponential kernel with 10 ms decay (as estimated from our multicompartmental model; Fig. 6A). Because the spontaneous activity of moth ORNs is very low (0.34(0.03) Hz in *A. ipsilon*[15]; 0.5–0.8 Hz in *S. littoralis*[45]), we neglected the activation of unbound receptors.

We fitted the transduction parameters to the first 400 ms after stimulus onset of the average LFP from the 20 and 200 ms stimuli. The model predicts well the time course of the firing rate in response to 20 ms, 200 ms, and 2 s stimuli and remarkably also to a more complex, time varying stimulus (Fig. 9). The fitted model parameters are

$k_a = 6.57 \times 10^{11}\,\text{s}^{-1}\,\text{M}^{-1}$, $s_a = 7.36\,\text{s}^{-1}$, $k_b = 37.3$, $s_b = 131\,\text{s}^{-1}$, and $\beta = -5.67\,\text{mV}$. The slow inactivation kinetics of the receptors, characterized by the time constant $\frac{1}{s_a} \approx 135\,\text{ms}$, is then responsible for the prolonged response to short stimuli.

## Prolonged response to short stimuli is maintained by the antennal lobe

ORNs project onto PNs and local neurons (LNs) within the AL. All ORNs expressing the same odorant receptor project their axons to the same glomerulus harboring the dendrites of PNs and LNs[46,47]. PNs create excitatory connections with other PNs. Most LNs provide inhibitory feedback both to PNs and LNs. PNs then project their axons to higher brain centers. Therefore, an understanding of how PNs reshape the firing response is essential for understanding the neural correlates of insect olfactory behavior. Even though the observation of the inhibitory phase in moth ORNs is novel, previous studies have observed the inhibitory phase in PN responses, despite

**Fig. 8 | Hyperpolarizing currents in the soma might affect the LFP shape. A** Input current to a multicompartmental ORN model. **B, C** Membrane potential in the soma without and with an adaptation current $I_{ad}$. In **B**, no adaptation current was involved ($I_{ad} = 0$). In **C**, the adaptation current was calculated so that the somatic membrane potential resembled the firing rate of the ORN. The adaptation current then changed the time course of the LFP (**D**). **E, F** To test whether the LFP shape is affected by firing activity-activated hyperpolarizing currents, we abolished the firing activity by injecting TTX into the neurons. The TTX-treated ORNs ($N = 17$) exhibited a similar LFP response shape as the control ORNs ($N = 12$), including a peak in deflection towards the end of the stimulus, indicating that this slow deflection was not caused by the firing activity.

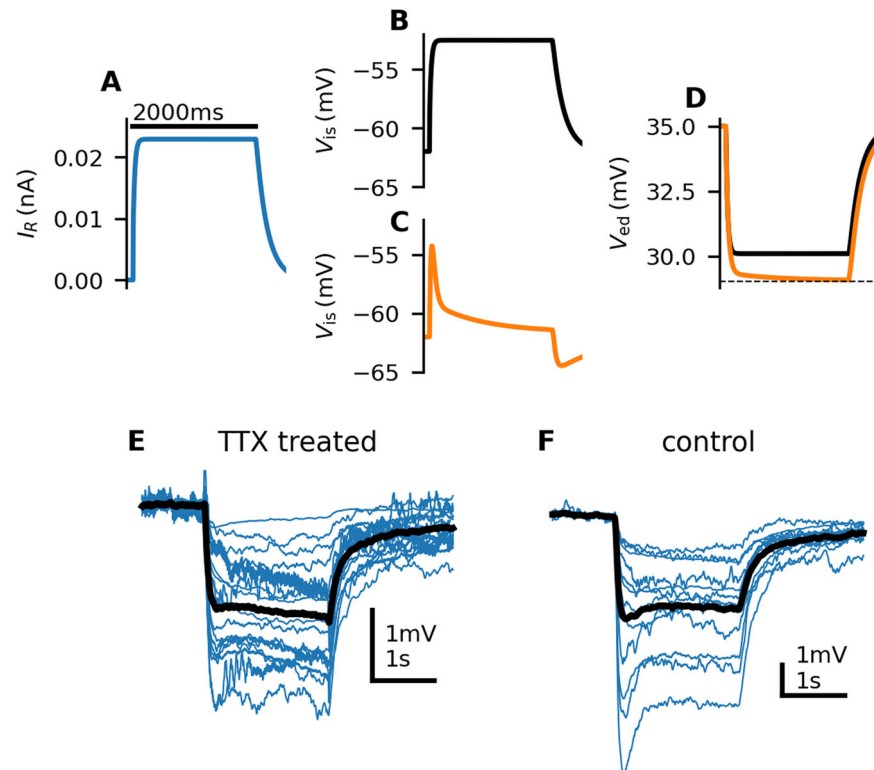

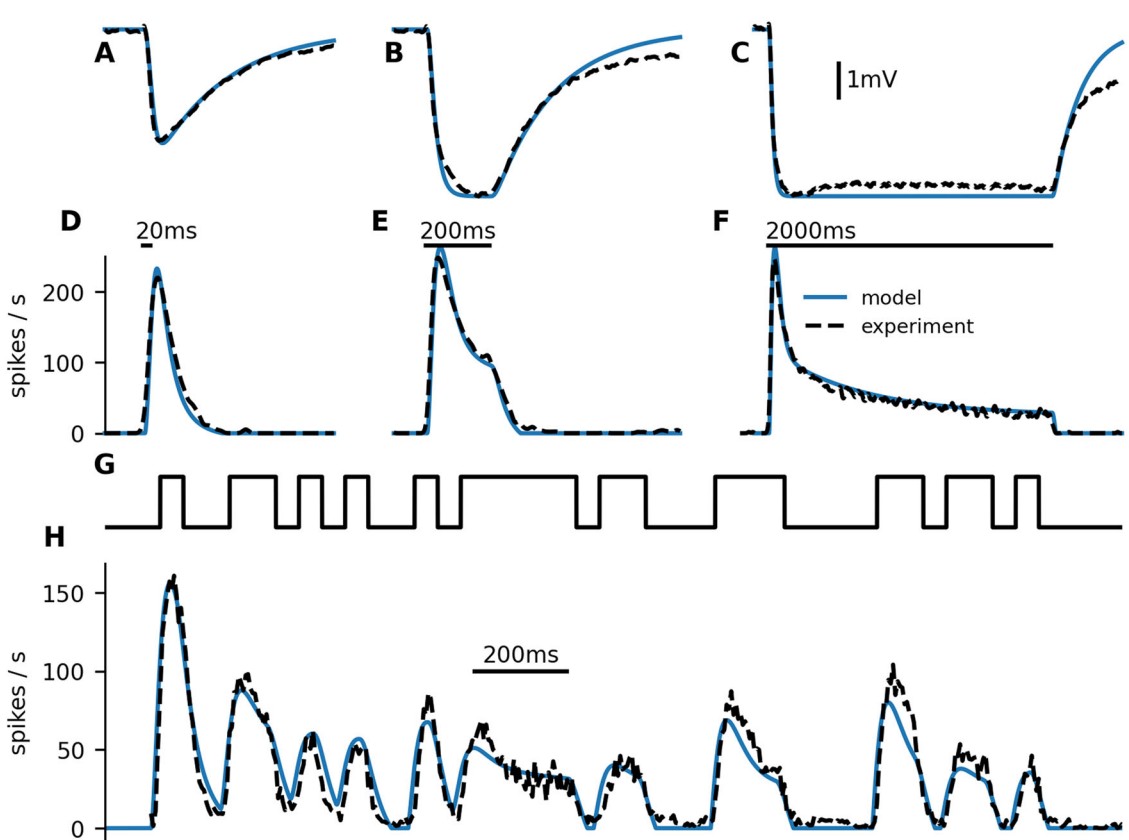

**Fig. 9 | Firing rate prediction an using odor transduction model. A–F** Prediction of LFP (**A–C**) and firing rate (**D–F**) using an odor transduction model (Eqs. (4) and (5)) combined with the linear-nonlinear model (Eqs. (1) and (2)). The transduction model was fitted to the average LFP (first 400 ms of the 20 ms and 200 ms stimuli), and the LN model was fitted to transform the average LFP to the average firing rate (2 s stimulus) (indicated by the dashed lines). Note that the model neglects receptor adaptation and the sustained activity. **G, H** Using the model to predict firing rate as a response to a complex stimulus. **G** Pheromone stimulus followed a frozen white noise pattern with a correlation time of 50 ms. **H** The model predicts the observed firing rate well. The predicted firing rate was multiplied by a factor of 0.77 to account for the lower concentration used in this experiment.

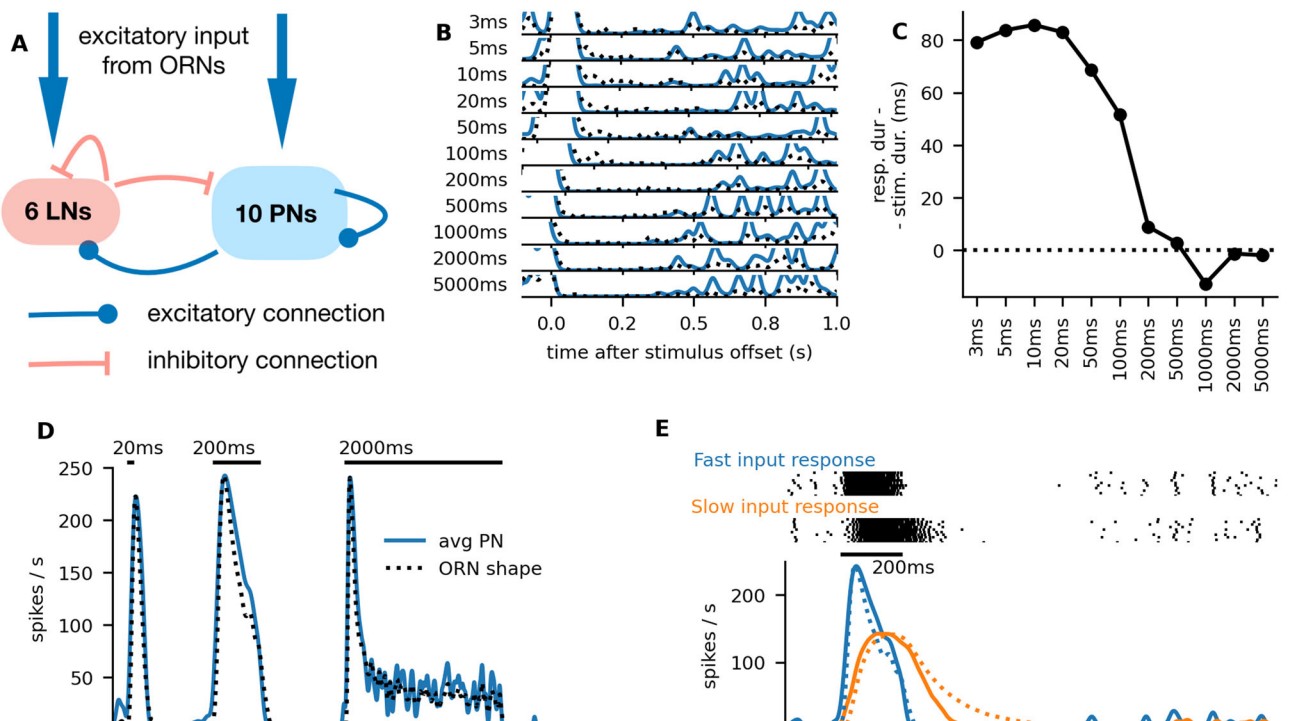

**Fig. 10 | Modeling the antennal lobe. A** Illustration of the used model. **B** The response end was clearly marked by an inhibitory phase, regardless of the stimulus duration (increasing from top to bottom, 3 ms to 5 s). The *y*-axis ranges from 0 to 20 Hz. Blue lines show the predicted PN responses. The ORN response is dotted and acts as an input into the AL model. **C** Although the inhibitory phase clearly marked the response end, the spiking response duration of the PNs still significantly exceeded the stimulus duration for stimuli shorter than 200 ms. **D** Average firing rates of the simulated PNs in response to stimuli of different durations. Dotted ORN firing rates were used as an input. Note that the ORN input firing rate is not to scale and is normalized to the peak of the PN firing rate for shape comparison. **E** The raster plots at the top show the spike trains of the 10 PNs in response to the unmodified ORN firing profile (Fast input) and the ORN firing profile smoothed with exponential kernel of 150 ms mean (Slow input). The PNs with the slow input also exhibited the inhibitory phase but did not track the stimulus duration. The full lines in the bottom panel show the PN firing rate averaged over 36 simulations. The dotted lines show the ORN input, rescaled for shape comparison.

using the classical odor-delivery device with a Pasteur pipette[15,48]. Moreover, PNs are sensitive to the slope of ORN firing rate[49], which can explain their transient responses. These results suggest that although ORNs are not obviously encoding the stimulus duration of short stimuli (Fig. 2), the ORN responses could be processed by the AL to provide a more accurate representation of the stimulus duration.

We used the ORN firing rates as an input to an AL model ([32,33]; see Materials and methods for details). We modeled a single glomerulus containing 10 PNs and 6 LNs. PNs created random excitatory connections to PNs and LNs within the glomerulus, and LNs created random inhibitory connections to PNs and other LNs (Fig. 10A). The PNs were equipped with small conductance $Ca^{2+}$-activated $K^+$ channels (SK channels), which together with the inhibitory input facilitated spike frequency adaptation and made the PNs sensitive to the slope of the ORN input. PNs exhibited a transient inhibition at the end of the stimulus, even when no transient inhibition was observed in the ORN response, in agreement with Jarriault et al.[15] (Fig. 10B). However, the response to short stimuli still significantly exceeded the stimulus duration (Fig. 10C) and the firing profile shape with this model did not differ greatly from the firing profile shape of ORNs (Fig. 10D). Therefore, we expect that the encoding of duration is not significantly altered by the antennal lobe and thus the longer responses to short stimuli likely propagate further and affect the behavioral responses.

PNs can exhibit the inhibitory phase even when there is no inhibitory phase in the ORN response[15]. Yet, we illustrate that their precision of stimulus duration encoding is improved by the observed dynamics in ORNs. We made the ORN response less sharp by convolving it with an exponential kernel of 150 ms mean. The smoothed ORN firing profile did not then show any inhibitory phase, but the inhibitory phase was clear in the PN responses.

However, the onset of the inhibitory phase occurred significantly later than the offset of the stimulus (Fig. 10E).

## Discussion
### Tri-phasic response of moth ORNs
The prolonged discharge of action potentials by ORNs in response to an odor stimulus, exceeding the duration of the stimulus, has been reported in many different species (locust[50], honeybee[51], cockroach[52], moths[14–17], some ORN-odor combinations in *Drosophila*[11,12], and *Drosophila* larva[53]). Cockroach ORNs can faithfully encode both the onset and offset of a stimulus by having a pair of ORNs in each sensillum, one sensitive to odor onset and another to odor offset[54,55]. A similar mechanism was reported in locust PNs[56], and recently also locust ORNs have been shown to display a multitude of patterns in response to odor stimulation[57]. However, such mechanism of paired responses has not been reported either for moth pheromone-sensitive ORNs or PNs. Instead, it has been shown that moth PNs terminate their response with an inhibitory phase, which helps encode the duration of the stimulus[15,58,59]. However, for stimulus duration shorter than 500 ms the response duration of the PNs still significantly exceeded the stimulus duration. In our study we showed that encoding of the stimulus duration happens very precisely already at the level of ORNs.

We found qualitative differences between the responses to short (<200 ms) and long (>200 ms) stimuli. Although the spiking response to a short stimulus exceeded the stimulus duration, the spiking response to a long stimulus ended with the stimulus. The response to long stimuli marked precisely the stimulus offset with an inhibitory phase. We showed that the prolonged response followed from the prolonged response of the LFP and could be explained by slow receptor inactivation kinetics, and the rapid

response termination was a result of the slow adaptation of the spike generator, which did not strongly affect the response to brief stimuli.

This inhibitory phase marking the end of stimulus has been observed previously with high volatility odors in *Drosophila*[10–12,49]. Moreover, we observed independence of the firing response shape on the odor dose, which was also reported in *Drosophila* with high volatility odors. It was predicted that independence on concentration and rapid response termination would also be observed with less volatile odors if delivered precisely[12]. We improved the odor-delivery device and brought experimental verification in the moth olfactory system. Although we are not aware of the qualitative differences in firing responses to short and long stimuli being reported for *Drosophila*, it has been shown that the LFP is significantly slower than the odor dynamics[11] and therefore the prolonged response to short stimuli can be expected as well. The newly observed similarities between *Drosophila* and moth ORNs unite the research in these different species.

On the other hand, we observed that the response pattern of *Drosophila* ORNs in the T1 sensilla to stimulation with the pheromone cVA differs qualitatively from the response of moth ORNs and *Drosophila* ORNs sensitive to VPCs. The response was tonic, in contrast to phasitonic. This difference can be compared to the difference in the purpose of the different ORNs. While moth pheromone sensitive ORNs and *Drosophila* VPC sensitive ORNs help with mid- to long-range navigation in turbulent environments, the cVA sensitive ORNs are involved in very short distance behaviors. This finding supports the idea that the phasi-tonic response evolved specifically to aid navigation in turbulent environments.

The inhibitory phase in moth ORNs was followed by a sustained increase in the firing activity long after the stimulus termination. We observed a sustained LFP more negative than the pre-stimulus level after the stimulus termination, indicating that the sustained firing activity was due to the sustained activity of the receptors. With classical odor-delivery devices, where the valve controlling the stimulus first leads into a glass mixing tube, such sustained activity could be explained by the slow release of pheromone molecules from the device surfaces after closing the valve that controls the stimulus. However, in our experiments, we strongly reduced the possibility of any pheromone molecules adhering to the odor-delivery device. The sustained activity could be caused instead by odor molecules adhering to the sensilla and/or it could represent an elevated probability of spontaneous OR-Orco channel opening after prolonged ligand-receptor interaction. Sensitization of ORNs was observed in *Drosophila* ORNs[60] and with heterologously expressed OR-Orco proteins[61]. This OR sensitization process requires Orco activity and was proposed to depend on cAMP production that would activate two feedback loops involving protein kinase and $Ca^{2+}$-calmodulin[62]. Regardless of the exact mechanism leading to the sustained activity, ORNs seem to remain slightly depolarized long after the stimulus termination, and their detection threshold is thus decreased. It is possible that ORNs evolved to have a very low spontaneous activity prior to any stimulation but after sufficient pheromone exposure can increase in activity to decrease the detection threshold and allow the ORNs to respond with higher intensity following a previous stimulus.

## Mechanism of spike frequency adaptation
We showed that the shape of the ORN's firing response can be captured very well with only two adaptation time scales: 31 and 635 ms. This is the minimal model capable of explaining the transiency of the firing response and the observed temporal resolution limits of the ORN. These adaptation time scales should correspond to time scales of the physiological processes responsible for the adaptation.

We suggested that the slow adaptation could be $Ca^{2+}$-dependent. A common $Ca^{2+}$ mechanism of spike frequency adaptation is hyperpolarization by $Ca^{2+}$-gated potassium channels, opening upon the $Ca^{2+}$ influx during an action potential. The hyperpolarizing currents in the soma should be reflected in the LFP. We hypothesized that the abolition of spiking activity would inactivate these currents and change the LFP shape. However, we did not observe any significant change in the LFP shapes of TTX-treated

neurons, indicating that if the adaptation is caused by hyperpolarizing currents, these currents are not dependent on firing activity.

Inactivation of voltage-gated sodium channels ($Na_V$) could also be responsible for the phasicity of the spiking response[11,63–65]. However, the timescales typical for inactivation (and reactivation) of $Na_V$ channels (4.8 ms measured in cultured honeybee ORNs[66]) were not needed to reproduce the firing rate profiles. It is, therefore, unlikely that the fast $Na_V$ channel inactivation contributes significantly to shaping the firing response. Some $Na_V$ channels also exhibit adaptation at slower time scales[67–71]. Moreover, in a recent study, a depolarization block of *Drosophila* ORNs was observed at high odorant concentrations and was shown to be likely caused by slow inactivation of $Na_V$ channels[72]. Patch clamp experiments on insect ORNs designed to measure the slow adaptation of $Na_V$ channels in insect ORNs would help to understand the physiological mechanisms behind their adaptation.

Our protocol for identifying the adaptation time scales can also be useful to investigate differences between species. Odor-to-firing rate or LFP-to-firing rate linear filters have been proposed in previous studies (e.g.,[11,12,73]). However, the filters were defined by their full time course, and, therefore, comparing them requires a comparison of the full time course of the filter instead of several interpretable parameters. Other works proposed an LFP-to-firing rate linear filter in *Drosophila* ORNs composed of two gamma distribution shaped functions with time constants 6 ms and 8 ms and with different signs to produce the bi-lobed shape of the filter[40,74]. The 8 ms adaptation time constant is relatively short compared to the 31 ms time constant we inferred for the moth ORNs and the *Drosophila* model does not include the slower adaptation time constant. The use of our time scale optimization protocol with *Drosophila* data would, therefore, help to better understand possible physiological differences in odor processing between the moth and *Drosophila* ORNs.

## Modeling the ORN response
We proposed a minimal model that links the stimulus to the firing rate, which captures well the firing profile of responses to isolated square pulses as well as to more complex stimuli. This model, due to having only a few parameters (five parameters for the transduction model and three parameters specifying the LFP-to-firing rate transformation) can be easily used to model the input to higher brain centers, which is otherwise often modeled as a piece-wise exponential function[32,33,75]. The following extensions to the model could be considered:

1. Adaptation of the odor receptors.
2. Persistent receptor activity.
3. Nonlinearity of the slow adaptation process.

Various receptor adaptation models were proposed for *Drosophila* ORNs[11,40,76], and we believe that these models could be also successfully applied to the moth ORNs. However, in the case of the moth, the long-lasting pheromone transduction pathway (due to pheromone adherence to the sensilla and/or sustained increased probability of spontaneous receptor opening) needs to be included as well to balance the adaptation and maintain receptor activity after the stimulus offset and to avoid transient LFP overshoot, as observed in some *Drosophila* ORNs[11]. It is also possible that the physics of fluid (air) movement across morphologically distinct antennal types (globular in *Drosophila*, feather-like in *A. ipsilon*), and the wingbeat frequency of the insect (200 Hz in *Drosophila*, 5–20 Hz in moths) that re-sculpt the odor plume could have both contributed to the evolution of the differentiated transduction process.

Our linear-nonlinear model predicts well the time course of the firing rate during stimulation and its offset after stimulus termination. However, the predicted duration of the inhibitory period is longer than what we generally observe. We believe that this can be explained by a voltage dependency of the slow adaptation process. Such non-linearity seems plausible, because either the $Na_V$ channels can recover faster at low membrane potential values, or the voltage dependency of the $Ca^{2+}$-gated $K^+$ channels causes them to close rapidly at membrane potentials below −40 mV[43].

## Implications for behavior and navigation efficiency

Male moths reach the pheromone source most reliably and with the least amount of counter-turning if the source is pulsating[5,77,78]. In particular, in experiments done with the almond moth *Cadra cautella*, the pulse duration was 130 ms, and the air-gap duration between pulses was 83 ms. These observations correlate well with our results showing that the ORNs exhibit prolonged firing in response to short (<200 ms) stimuli. Moreover, prolonged response to very short stimuli (e.g., 3 ms) can ensure that the brief encounter is registered by the brain and can be acted upon Lei et al.[79] showed that the ability of the antennal lobe to track stimuli is essential for maintaining an upwind flight. In their protocol, the moth encountered odor filaments with mean frequency between 3.8 and 4.47 Hz, leading to 224 ms average inter-pulse interval in the latter case. It is not yet determined which component of the plume-tracking behavior is correlated with what phase of the responses but we hypothesize that inhibitory periods are necessary for the transition from upwind flight to casting behavior. With very brief stimuli, this would suggest that moths continue in an upwind flight for some period of time, even after PNs (and presumably also ORNs) stop firing. It remains to be seen what is the duration of silence of the PNs that triggers casting behavior, possibly by testing the lowest frequency of odor encounters at which the moth still maintains an upwind flight.

A study on *Drosophila* showed that the switch from casting to upwind flight lags approximately 190 ms behind the encounter of the odor plume, while the lag between leaving the odor plume to switching to casting from upwind flight is approximately 450 ms. Analysis of the dependence of the lag on the duration of how long the fly stays in the plume could reveal whether the lag is longer for brief encounters. Such result would confirm our hypothesis that the duration of the upwind flight depends on the duration of the ORN response, rather than duration of the stimulus.

On the other hand, the slow (635 ms) adaptation allows the moth to respond rapidly to a loss of pheromone signal after prolonged exposure, but possibly also to adapt to the background intensity within a pheromone plume.

## Materials and methods

### Insects

*A. ipsilon* and *S. littoralis* adult males were fed an artificial diet[80]. Pupae were sexed and males and females were kept separately in an inversed light-dark cycle (16 h:8 h light:dark photoperiod) at 22 °C. Experiments were carried out on 5-day-old males.

*Drosophila* were reared on standard agar-cornmeal diet and maintained at 25 °C in 12:12 light–dark conditions. Recordings were conducted on 3–4-day-old *Drosophila* males.

### Chemicals

The main components of the sex pheromones of *A. ipsilon* (Z7-12:Ac; CAS 14959-86-5), *S. littoralis* (Z9,E11-14:Ac; CAS 50767-79-8), and *Drosophila* pheromone 11-*cis*-vaccenyl acetate (cVA; CAS 6186-98-7) were bought from Pherobank (purity > 99%). Linalool (CAS 78-70-6; purity > 97%), α-pinene (CAS 80-56-8; purity > 98%), acetone (CAS 67-64-1), and (Z)-3-hexenyl acetate (CAS 3681-71-8; purity > 95%) were bought from Sigma-Aldrich. They were diluted at 10% in mineral oil (CAS 8012-95-1).

### Odor delivery

Analysis of the dynamics of odor coding requires either monitoring or controlling the temporal resolution of odor stimuli. Monitoring the odor stimulus can be done with a PID with high temporal resolution[3]. Unfortunately, common moth pheromones cannot be detected by a PID, because their ionization energies are too high for the PID lamp. Proton transfer reaction-mass spectrometers (PTR-MS) can monitor the dynamic of odor plumes[81], including pheromone plumes. However, the sensitivity of PTR-MS remains too low to monitor pheromone stimuli at physiological concentrations. Therefore, we developed a new odor-delivery device to better control the stimulus dynamics.

Our odor-delivery device was based on two serially connected electrovalves. The first electrovalve (any of EV1–EV8, further referred to as an upstream valve) odorized the passing airflow. The second electrovalve (EV9, downstream valve) controlled the timing of the stimulus (Fig. 11).

The airflow running into the odor-delivery device was regulated to 2.5 bar with a pressure regulator (Numatics 34203065, Michaud Chailly, Voisins-le-Bretonneux, France) coupled to a 25 µm filter (Numatics 34203065). The incoming air was charcoal filtered (hydrocarbon trap, Cat. #22013, Restek, Lisses, France) and humidified by passing in through a bottle with distilled water, except for experiments with PID. The airflow was then divided into 8 flows (200 mL/min each) with an airflow divider (LFMX0510528B, The Lee Company, Westbrook, CT, USA). Each of the 8 flows was connected to a 3-way electrovalve (EV1 to EV8; LHDA1223111H, The Lee Company). Normally open (NO, non-odorized) and normally closed (NC, odorized) exits of the 8 valves were connected to a low dead-volume manifold (MPP-8, Warner Instruments, Holliston, MA, USA) or to odor sources, respectively. The non-odorized airflow permanently bathed the insect preparation. All outlets of odor sources were connected to a second MPP-8 manifold that was connected to an electrovalve (EV9; LHDA1233215H, The Lee Company). The NO exit of EV9 was vented into the vacuum. A small glass tube (10 mm total length, 1.16 mm internal diameter, resulting in airspeed of 3.2 m/s) bent at 90° and connected to the EV9 NC exit facilitated focusing the stimuli on the insect antenna. EV9 and the small bent tube were thus the sole surfaces on which odor puffs controlled by EV9 could adsorb and thus alter the stimulus dynamics. The outlet of the small tube was positioned under the dissecting microscope at 1 mm from the recorded sensilla. An aluminum shield connected to the ground around EV9 minimized artifacts during the opening and closing of the valve. The downstream part of the odor-delivery device (from the manifold to EV9 and the attached small bent tube) was decontaminated after each experiment in an oven for 60 min at 80 °C with an airflow injected from the small bent tube with EV9 activated. All tubing but the one that delivered the permanent airflow was made of Teflon (internal diameter 1.32 mm). The shape of stimuli delivered to the antenna was measured using a mini PID (Aurora Scientific Inc, Aurora, Canada).

### Equilibration and stability of the odor source

After opening the upstream electrovalve two processes are at play when the airflow passes through the odor source, with opposite effects on the concentration of odor reaching the downstream electrovalve, EV9.

1. Dilution of the head-space, which reduces the concentration of odor delivered to EV9. This has an effect that increases with time until an asymptote is reached corresponding to the equilibrium of odor molecules desorbing from the filter paper and those carried out of the vial by the airflow.
2. Reversible binding of odor molecules to the surfaces of the odor-delivery device, which reduces the concentration of odor delivered to EV9. This has an effect that gradually decreases over time until it becomes null when the adsorption/desorption equilibrium is reached.

Using linalool (diluted at 10%) and the PID, we verified how long the upstream valve must be open before the odor concentration delivered to the downstream valve becomes constant (further referred to as equilibration time). With no or short equilibration times (≤2 s), PID responses were not square but had a decreasing amplitude indicating that the dilution of head-space was dominant. When the equilibrium time was at least 10 s, the PID response to a 0.5 s stimulus was square. Increasing the equilibration time to more than 10 s had very little effect on the amplitude of the PID response (Fig. 11B, C). We kept the same 10 s equilibration time when using α-pinene and acetone, which are more volatile than linalool.

Because the PID cannot monitor pheromone stimuli, the equilibration time with pheromone was adjusted by measuring the amplitude of single sensillum recording responses to a 0.5 s stimulus with 100 pg of Z7-12:Ac. Equilibration times of 1, 3, 10, 30, and 78 s were tested both in ascending and descending order. Stimuli were applied every 2 min. Equilibrations were stopped at each stimulus offset. The amplitude of responses increased for equilibration times of 1–30 s and then remained stable, indicating that the

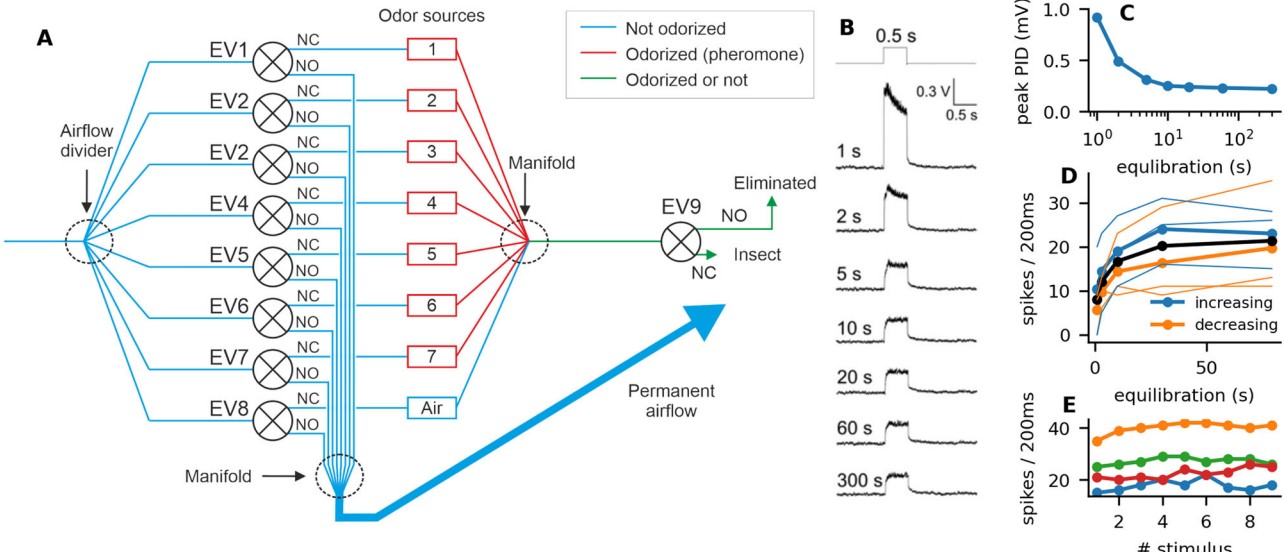

**Fig. 11 | Odor-delivery device and its equilibration. A** Schematics of the developed odor-delivery device. NO normally open (no stimulus) and NC normally closed (during stimulus). The insect was placed at 1 mm after EV9. **B–E** Testing of equilibration times and source stability. **B** PID responses to 0.5 s-linalool stimuli with different equilibration times. When the equilibration was too short, the PID response exhibited a transient peak. **C** With an equilibration of approximately 10 s the peak was no longer present, and the amplitude of the response did not change significantly with longer equilibration times. **D** Number of spikes recorded during 200 ms in response to 100 pg of Z7-12:Ac for different equilibration times. Each ORN was presented with 5 stimuli with different equilibration times (1, 3, 10, 30, and 79 s) either in increasing or decreasing order. For each order, the line is an average of 3 ORNs. The black line is the average of all 6 ORNs. **E** We measured the stability of the pheromone source first by applying 9 stimuli with 0.1 ng of Z7-12:Ac. Each stimulus was preceded by an equilibration time of 30 s. The inter stimulus interval was 2 min. Each line represents the response of a single ORN.

odor binding to surfaces was the dominant effect (Fig. 11D). We thus kept an equilibration time of 30 s for further experiments with Z7-12:Ac.

We then measured the stability of the pheromone source by applying 9 stimuli with 100 pg of Z7-12:Ac. Each stimulus was preceded by an equilibration time of 30 s. The inter-stimulus interval was 2 min. The amplitude of responses remained constant over the 9 stimuli (Fig. 11E). Thus, responses remained stable after $9 \times 30$ s = 4.5 min of equilibration with the upstream valve.

## Single sensillum recordings

For single sensillum recordings on male moths, the insects were briefly anesthetized with $CO_2$ and restrained in a Styrofoam holder. One antenna was immobilized with adhesive tape. In *A. ipsilon*, single sensillum recordings were carried out from the long trichoid sensilla located along antennal branches, the vast majority of which house an ORN tuned to Z7-12:Ac[15] and in *S. littoralis* from the long trichoid sensilla housing an ORN responding to Z9,E11-14:Ac[45]. We used either tungsten electrodes or glass electrodes, the latter of which allowed recording of the local field potential (LFP) in addition to the firing response of ORNs. In both cases, the recording electrode was inserted at the base of a long trichoid sensillum. The reference electrode was inserted in an antennal segment next to the one bearing the recorded sensillum. In the majority of recordings (about 90%), only one spike amplitude was present in the recording. When two amplitudes were visible, only the higher amplitude was used (see also Jarriault et al.[15]).

For single sensillum recordings on *Drosophila*, 3- to 4-day-old flies were restrained inside a 200 µL pipette tip with only the head protruding from the narrow end. The pipette tip was fixed with dental wax onto a microscope glass slide with the ventral side of the fly facing upward. Then, the antenna was gently placed on a piece of a glass slide and maintained by placing a glass capillary between the second and third antennal segments, held in place by dental wax. The slides were placed under a light microscope (BX51WI, Olympus, Tokyo, Japan) equipped with an ×100 magnification objective (SLMPlan 100×/0.60, Olympus) and ×10 eyepieces. Recordings were carried out using tungsten electrodes, with the recording electrode inserted in T1 sensilla containing a cVA-responding ORN[82] and the

reference electrode inserted in one eye. Recordings were done in response to cVA stimuli from wild-type flies as well as in response to Z7-12:Ac from flies with the genotype *w*; *UAS-AipsOr3*; *Or67d*$^{GAL4[2]}$ expressing the AipsOR3 receptor sensitive to Z7-12:Ac instead of the endogenous receptor Or67d[41].

Recordings were done using a CyberAmp 320 controlled by pCLAMP10 (Molecular Devices, San Jose, CA, USA). The signal was amplified (×100), band-pass filtered (10–3000 Hz) with tungsten electrodes or low-pass filtered (3000 Hz) with glass electrodes and sampled at 10 kHz with a Digidata 1440A acquisition board (Molecular Devices). Spikes were sorted using Spike 2 software (CED, Oxford, Great Britain; version 10.07).

Typically, we recorded from 1 to 3 sensilla from each animal.

## Experimental protocols

To record the firing responses to pulses of different durations (Fig. 2), we performed recordings with tungsten electrodes from 23 sensilla and presented them with stimuli of durations 3 ms, 5 ms, 10 ms, 20 ms, 50 ms, 100 ms, 200 ms, 500 ms, 1 s, 2 s, and 5 s (pheromone dose 100 pg) in a randomized order. There was a 2 min gap between stimuli. The number of recorded responses varied for each duration and is provided in Table 1.

To test the responses to different pheromone doses (Fig. 3), we performed recordings with tungsten electrodes from 57 sensilla, presenting them with pulses of durations 20 ms, 200 ms, and 2 s in a randomized order, but with an increasing pheromone dose. The number of responses recorded for each duration-dose pair is shown in Table 2.

We recorded the LFP simultaneously with the spiking activity for the pulse durations 20 ms, 200 ms, and 2 s, presented in a randomized order with 3 min inter-stimulus intervals (dose 1 ng). To exclude neurons whose functioning was altered, we presented an additional 2 s pulse after the initial three pulses and included the recording in the analysis only when the second response to the 2 s pulse exhibited the inhibitory phase. In total, we used 26 out of 37 ORNs, therefore 26 responses for each duration. To filter out the action potentials from the LFP, we used a 15 Hz 2-pole Butterworth low-pass filter.

In experiments with intermittent stimuli, a noise sequence was generated by randomly opening/closing the odor-delivery valve (EV9) every

**Table 1 | Number of sensilla recorded for each pulse duration**

| pulse duration: | 3 ms | 5 ms | 10 ms | 20 ms | 50 ms | 100 ms |
|---|---|---|---|---|---|---|
| | 22 (7) | 22 (13) | 23 (21) | 22 (20) | 21 (20) | 23 (22) |
| pulse duration: | 200 ms | 500 ms | 1 s | 2 s | 5 s | |
| | 23 (22) | 23 (23) | 23 (22) | 23 (22) | 23 (22) | |

The number of neurons that responded by firing at least 5 spikes in the first 100 ms after stimulus onset is shown in brackets.

**Table 2 | Number of sensilla recorded for each duration-dose pair**

| | 20 ms | 200 ms | 2 s |
|---|---|---|---|
| 10 pg | 57 (28) | 57 (32) | 57 (32) |
| 100 pg | 55 (33) | 56 (44) | 54 (38) |
| 1 ng | 53 (40) | 52 (39) | 52 (41) |

The number of neurons that responded by firing at least 5 spikes in the first 100 ms after stimulus onset is shown in brackets.

50 ms during a 2 s interval. Each ORN was stimulated with the generated sequence every 30 s, switching between intermittent (dose 20 pg) and constant (dose 10 pg) stimulus, up to 40 times (20 times for each stimulus; the first sequence was either constant or intermittent). Sometimes ORNs started responding more tonically with time, no longer exhibited the inhibitory period after 2 s, and could not follow the intermittent stimulus. We assume that this was due to damage to the ORN, and we discarded the recordings from the moment the ORN stopped exhibiting the inhibitory phase after the 2 s constant stimulus (the inhibitory phase here defined as zero spikes during the interval 50–500 ms after the stimulus offset). Altogether, we used 554 different responses to the intermittent stimulus from 55 different ORNs.

For the experiments using TTX, the drug was dissolved (50 μM) in saline (in μM: NaCl 154, KCl 3, glucose 24) and injected into the body of the antenna using a syringe-driven glass micropipette. Controls were saline injections. Recordings started 5 min after injection. The firing activity was completely abolished after all TTX injections and remained intact after the saline injections.

## Data analysis

We estimated the firing rates by the kernel density estimation method. Each spike was substituted with a normal distribution probability distribution function with mean at the spike time and standard deviation $\sigma = \frac{bw}{2}$, where bw is the kernel width.

In Fig. 2, we used a time-dependent kernel width to depict the responses to short stimuli with sufficient detail but avoided high noise when the firing rate dropped during longer stimulation. The time dependence was given by:

$$bw(t) = \begin{cases} bw_{min} & t < 0, \\ bw_{max} - bw_{min} \exp(-t/\tau_{KDE}) + bw_{min} & t > 0, \end{cases} \quad (6)$$

where $bw_{min} = 10$ ms, $bw_{max} = 100$ ms, $\tau_{KDE} = 500$ ms, and we assumed that the stimulus onset is at 0.

The first inter-spike interval (ISI) that finishes after stimulus offset and exceeds 100 ms is considered as the terminating ISI and the initiating AP as the time of the response end. We calculated the response end only if the neuron fired at least 5 action potentials during the first 100 ms after the stimulus onset (the numbers of responding neurons are provided in brackets in Table 1 and Table 2). We then calculated the time of the response end for a group of neurons as the median of individual response ends (red vertical lines in Fig. 2 and Fig. 3). Note that if the ISI after the last spike

during stimulation is longer than 100 ms, the calculated response end for the ORN is before the stimulus end.

## Linear-nonlinear model for firing rate prediction

We used linear regression to predict the firing rate. As independent variables, we used values of the past LFP convolved with a gamma distribution-shaped function with different time constants and shape parameters:

$$x(t; \tau, \alpha) = \int_0^{+\infty} V(t-s) \frac{1}{\Gamma(\alpha)\tau^\alpha} s^{\alpha-1} e^{-\frac{s}{\tau}} ds, \quad (7)$$

where $V$ is the LFP. The model is then specified by the time constants $\tau_1, \ldots, \tau_n$ and the corresponding shape parameters $\alpha_1, \ldots, \alpha_n$. The estimated firing rate before the non-linearity is specified by the coefficients $c_1, \ldots, c_n$:

$$f(t) = \sum_{k=1}^{n} c_k x(t; \tau_k). \quad (8)$$

We estimated the coefficients using the least square method to provide an estimate of firing rate (estimated from the spike train with kernel width 30 ms) during the 2 s stimulus and 1 s of the preceding spontaneous activity.

In order to choose the optimal $(\alpha, \tau)$ pairs, we first used a model with 20 time constants, ranging from 1 ms to 3 s, equidistantly spaced on the logarithmic scale and 17 different gamma distribution shapes $\alpha$ ranging from 1 to 5, equidistantly spaced. The model then contained $20 \times 17$ independent variables. We fitted the model to the average LFP and average firing rate response during 2 s stimulation with lasso regression (the optimal L1 penalty parameter was selected with cross-validation using the LassoCV regressor in Scikit-learn[83]).

The non-zero coefficients then concentrated around several $(\alpha, \tau)$ pairs, but mostly at three areas at the $\alpha = 1$ edge, which led us to select three different $(\alpha, \tau)$ pairs as initial values for optimization: (1, 1 ms), (1, 40 ms) and (1700 ms). We then optimized the time constants using the L-BFGS-B algorithm implemented in SciPy[84] (we kept $\alpha = 1$, because allowing $\alpha > 1$ resulted only in a negligible improvement). The lower bound for the shortest time constant was set to 0, where we defined the kernel with $\tau = 0$ as a $\delta$-function, resulting in an unfiltered value of the LFP. We found that the firing rate can be reliably predicted from the LFP using only two time scales: 31 ms and 635 ms, and the unfiltered LFP.

## Modeling odor transduction

We modeled the transduction described by Equation (4) by a set of differential equations:

$$\frac{d}{dt} R = s_b \cdot OR - [O]k_b s_b \cdot R, \quad (9)$$

$$\frac{d}{dt} OR = [O]k_b s_b \cdot R + s_a \cdot OR^* - k_a s_a \cdot OR - s_b \cdot OR, \quad (10)$$

$$\frac{d}{dt} OR^* = -s_a \cdot OR^* + k_a s_a \cdot OR, \quad (11)$$

$$\frac{d}{dt} LFP = -\frac{1}{\tau_{LFP}} (LFP - \beta \cdot OR^*). \quad (12)$$

R, OR, and OR* indicate the ratios of unbound, bound, and activated bound receptors, $\tau_{LFP} = 10$ ms. The initial conditions were R = 1 and OR = OR* = LFP = 0. We modeled the odor concentration as a square odor pulse: $[O] = 10^{-11}$ M during stimulation and 0 otherwise. Because we did not attempt to model the adaptation or the sustained activity (more important with long stimuli), we fitted the parameters $s_b, k_b, s_a, k_a$, and $\beta$ to the first 400 ms after stimulus onset of the average LFP from the 20 ms and 200 ms stimulations. We fitted the parameters by minimizing the square error of the prediction with the L-BFGS-B algorithm implemented in SciPy[84].

## Table 3 | Synaptic connection amplitudes

|     | $S_{exc}$ | $S_{inh}$ | $S_{slow}$ | $S_{stim}$ |
|-----|-----------|-----------|------------|------------|
| PN  | 0.01      | 0.0169    | 0.0338     | 0.004      |
| LN  | 0.006     | 0.015     | 0.04       | 0.0031     |

## Table 4 | Neuron connection probabilities

| PN → PN | PN → N | LN → PN | LN → LN |
|---------|--------|---------|---------|
| 0.75    | 0.75   | 0.38    | 0.25    |

### Antennal lobe model

We used a model of a single glomerulus from the antennal lobe (AL) model proposed by Tuckman et al.[32]. In the following, we explicitly state when we deviated from the established model.

The glomerulus contained 10 PNs and 6 LN. The membrane potential dynamics of $i$-th PN and $j$th LN were governed by the following dynamics:

$$
\begin{aligned}
\frac{d}{dt} V_{PN}^i = & -\frac{1}{\tau_V}(V_{PN}^i - E_L) - g_{SK}^i(t)(V_{PN}^i - E_{SK}) \\
& - g_{stim}^i(t)(V_{PN}^i - E_{stim}) \\
& - g_{exc}^i(t)(V_{PN}^i - E_{exc}) - g_{inh}^i(t)(V_{PN}^i - E_{inh}) \\
& - g_{slow}^i(t)(V_{PN}^i - E_{inh}),
\end{aligned}
\tag{13}
$$

$$
\begin{aligned}
\frac{d}{dt} V_{LN}^j = & -\frac{1}{\tau_V}(V_{LN}^j - E_L) - g_{stim}^j(t)(V_{LN}^j - E_{stim}) \\
& - g_{exc}^j(t)(V_{LN}^j - E_{exc}) - g_{inh}^j(t)(V_{LN}^j - E_{inh}) \\
& - g_{slow}^j(t)(V_{LN}^j - E_{inh}),
\end{aligned}
\tag{14}
$$

where $\tau_V$ is the membrane time constant, $g_{SK}$ is the conductance of SK channels, $g_{stim}$ is the excitatory conductance associated with the ORN input, $g_{exc}$ is the excitatory synaptic conductance from PNs, $g_{inh}$ is the fast inhibitory GABA$_A$ conductance, and $g_{slow}$ is the slow GABA$_B$ conductance. $E_{SK}$, $E_{stim}$, $E_{exc}$, and $E_{inh}$ are the reversal potentials associated with these conductances, $E_L$ is the leak reversal potential. The reversal potentials are expressed in nondimensional units: $E_L = 0$, $E_{exc} = E_{stim} = \frac{14}{3}$, $E_{SK} = E_{inh} = -\frac{2}{3}$. A neuron fires a spike when the membrane potential $V$ reaches the threshold $V_{thr} = 1$ and is then reset to $E_L$ and held at $E_L$ for $\tau_{ref}$. The synaptic conductances $g_X$, $X \in \{exc, inh, slow, stim\}$ follow the equation

$$
\tau_X \frac{d}{dt} g_X^i = -g_X^i + S_X \sum_{t_{spike} \in \{t_X^i\}} \delta(t - t_{spike}),
\tag{15}
$$

where $\{t_X^i\}$ represents the corresponding presynaptic spikes to the $i$th, $\tau_X$ is the synaptic time constant for the given synapse type and the conductance increases by $\tau_X S_X$ with each presynaptic spike arrival. $S_X$ differ for PNs and LNs and are specified in Table 3.

The SK conductance $g_{SK}$ was modeled only for the PNs and did not rise instantaneously but instead followed the equations:

$$
\tau_{rise} \frac{d}{dt} g_{SK}^i = -(g_{SK}^i - z),
\tag{16}
$$

$$
\tau_{SK} \frac{d}{dt} z = -z + S_{SK} \sum_{t_{spike} \in \{t^i\}} \delta(t - t_{spike}),
\tag{17}
$$

where $\tau_{rise}$ characterizes the rise time, $\tau_{SK}$ is the decay time constant of the SK conductance, and $\{t^i\}$ is the set of spikes fired by the $i$-th PN. Note that here, for simulation purposes, we deviated from the original model[32] by

modeling $g_{SK}^i$ with a set of two equations instead of modeling the time course of $g_{SK}^i$ following a single spike as a piece-wise function. $S_{SK}^i$ was drawn from a normal distribution with mean $\mu = 0.5$ and $\sigma = 0.2$ (negative values were set to 0). The connections between the neurons within the glomerulus were random with probabilities specified in Table 4.

Each PN and LN was set to receive a background ORN input of 3 spikes/ms. The time course of the additional stimulus coming from the ORNs during stimulus was given by the average ORN firing rate (Fig. 2, note that the input, therefore, differed from[32]) and was scaled to mimic the input from 100 ORNs (peak of approximately 25 spikes/ms). We simulated the network using the Brian 2 Python package[85].

### Multicompartmental ORN model

The model is a simplified version of the moth pheromone transduction model by Gu et al.[86]. From this model, we kept the morphology and the passive conductances (Fig. 12). The following set of equations describes the evolution of the potentials in the individual compartments:

$$
\begin{aligned}
\frac{dV_{id}}{dt} = & \frac{G_e}{C_{md}(G_e + G_i)}(I_R + I_{ld} - I_e) \\
& + \frac{G_e}{C_{ma}(G_e + G_i)}(I_a - I_e) \\
& + \frac{G_i}{C_{ms}(G_e + G_i)}(I_i - I_{ls} - I_{ad}),
\end{aligned}
\tag{18}
$$

$$
\begin{aligned}
\frac{dV_{ed}}{dt} = & \frac{G_i}{C_{md}(G_e + G_i)}(I_e - I_R - I_{ld}) \\
& + \frac{G_e}{C_{ma}(G_e + G_i)}(I_a - I_e) \\
& + \frac{G_i}{C_{ms}(G_e + G_i)}(I_i - I_{ls} - I_{ad}),
\end{aligned}
\tag{19}
$$

$$
\frac{dV_{is}}{dt} = \frac{I_i - I_{ls} - I_{ad}}{C_{ms}},
\tag{20}
$$

$$
\frac{dV_{ea}}{dt} = \frac{I_a - I_e}{C_{ma}}.
\tag{21}
$$

Where the currents are described by:

$$
I_{ls} = G_{ls}(V_{is} - E_{ls}),
\tag{22}
$$

$$
I_{ld} = G_{ld}(V_{ed} - V_{id} + E_{ld}),
\tag{23}
$$

$$
I_i = G_i(V_{id} - V_{is}),
\tag{24}
$$

$$
I_a = -G_a(V_{ea} + E_a),
\tag{25}
$$

$$
I_e = G_e(V_{ea} - V_{ed}).
\tag{26}
$$

$I_R$ is the receptor current, which we either calculated by fixing the LFP ($V_{ed}$) and calculating what receptor current $I_R$ is necessary to produce a given LFP time course, or we fixed the $I_R$ time course. To estimate $I_R$ from a given LFP, we substituted Eq. 19 with the numerical derivative of the LFP and expressed $I_R$ using the numerical derivative to use in Eq. 18.

$I_{ad}$ is the adaptation current. We considered $I_{ad} \neq 0$ only to illustrate the effect of adaptation currents in the soma on the LFP. In such case, we fixed the input $I_R$ to the model and fixed the time course of the somatic membrane potential $V_{is}$ to correspond to the shape of the firing rate (again, by calculating its numerical derivative and eliminating Eq. 20). Then, we calculated the necessary $I_{ad}$ to balance the depolarizing effect of $I_R$.

**Article**

**Fig. 12 | Schematic illustration of the multi-compartmental ORN model.** The model is composed of five different compartments: hemolymph, where the reference electrode is placed, auxiliary cells, sensillar lymph, dendrite, and soma. The difference of potential between the outer dendrite and the recording electrode ($V_{ed}$) corresponds to the extracellularly measured signal.

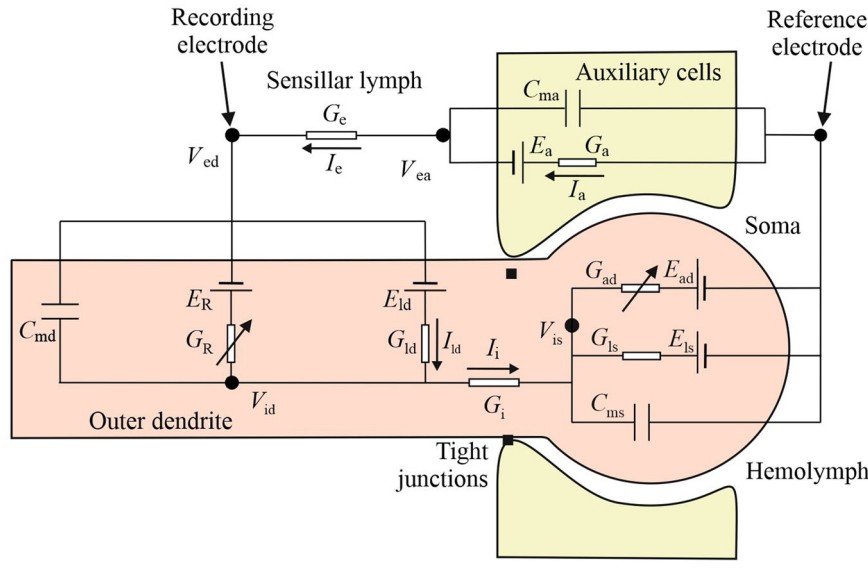

We simulated the multicompartmental model with the explicit Runge-Kutta method of order 5(4) with an upper limit on integration step of 0.1 ms implemented in SciPy[84]. We used the initial conditions $V_{ed} = V_{ea} = -35$ mV, $V_{id} = V_{is} = -62$ mV. This condition corresponds to $I_e = I_{ld} = I_i = I_{ls} = I_a = 0$, given that $I_R = I_{ad} = 0$.

### Statistics and reproducibility
Employed statistical tests and number of number of replicates are indicated per experiment in the Methods section, figure captions, and in the respective parts of the "Results" section.

### Data availability
Associated raw data are available in a Zenodo repository[87]. Numerical source data for all graphs in the paper can be found in Supplementary Data 1 file.

### Code availability
Associated Python code and Jupyter notebooks are available in a Zenodo repository[87].

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

## Acknowledgements

We are grateful to Claude Collet for a helpful discussion on sodium channels from insect ORNs and to Vincent Jacob for the critical reading of the manuscript. We thank Nicolas Montagné and Emmanuelle Jacquin-Joly for providing us with the *w*; *UAS − AipsOr*3; *Or*67*d*$^{GAL4[2]}$ flies. This work was supported by the Charles University, project GA UK No. 1042120, and the Czech Science Foundation project 20-10251S. TB benefited from a fellowship from the Plant Health and Environment Division of INRAE.

## Author contributions

Experiments were designed by P.L with contributions from T.B., C.M., E.D., A.C., and L.K. P.L. designed the odor delivery device. C.M. and E.D. performed the majority of the experiments. T.B. analyzed the experimental data, designed the mathematical models, and wrote the manuscript. P.L., L.K., and A.C. participated in writing and editing the manuscript. P.L. and L.K. supervised the research.

## Competing interests

The authors declare no competing interests.
