## [Peer Review File · Communications Biology]

Reviewers' comments:

Reviewer #1 (Remarks to the Author):

In their manuscript “Stimulus duration encoding occurs early in the moth olfactory pathway” the authors examine a potentially interesting question: how the moth olfactory receptor neurons (ORNs) encode varying the durations of odor filaments, which are likely similar to what the animal would encounter while navigating through an odor plume. The authors made physiological recordings from ORNs in two moth species and from *Drosophila*, and then based on these recordings, created a compartmentalized model ORN to examine the mechanism that drives stimulus duration encoding.

The authors clearly did a lot of work, but my enthusiasm is reduced for several reasons. The manuscript is quite difficult to follow and is missing some basic information needed to understand the results. This work includes some interesting ideas about mechanisms of adaptation at the level of signal generation, but, because the authors dramatically oversimplified the sensillum to include only a single ORN and disregarded the fact that ORNs are diverse and respond in different ways, it is difficult to know whether these ideas are physiologically relevant. Finally, the authors omitted from discussion and citation some very similar research.

Major Concerns:

1) Misrepresentation of the ORN responses. The authors recorded responses of moth ORNs to pulses of pheromones delivered in durations ranging from 3ms to 5s. But their figures show they inaccurately describe their own results and then base their model on these inaccuracies. For example, the repeated statement that short duration stimuli elicit prolonged responses while long stimuli do not, doesn't seem to be true. A more accurate description would be that pheromones elicit a brief burst of activity upon stimulus onset.

The authors did not justify their choices of analysis windows for inhibitory and rebound responses, and did not explain why there are gaps in between the stimulus offset and these defined windows. The authors further claim that ORN responses can be explained by two opposing forces, but do not explain the reasoning behind or significance of this claim. Overall, the representation of the data is very confusing and seemingly at odds with the results.

2) Lack of novelty in adaptation work. Kim et al 2023 showed that, in locusts, ORNs undergo different types of adaptation and describe how they affect responses to plume-like odor pulses.

The authors should cite and describe this closely-related work.

3) Oversimplification of ORN model. One of the manuscript's main points is that ORNs can encode stimulus duration. The authors focus on the structure of the ORN and a possible mechanism that drives different types of adaptations. However, the model includes little to no detail about the structure of the sensillum or the relationships between ORNs. Similarly, the model ignores distinctions among different types of ORNs that are recorded in a sensillum. The ORNs are agnostically treated as identical units, but previous studies in various species (Hallem and Carlson 2006, Raman et al 2010, Kim et al 2023) have demonstrated the great heterogeneity in ORN responses. Because the model oversimplifies the ORN responses it is difficult to know whether its conclusions are physiologically relevant.

4) Lack of clarity in *Drosophila* data. An experiment with wild type and mutant *Drosophila* is shown to illustrate "phasi-tonicity" is not receptor dependent. However, the experiment is missing an essential set of negative controls: Z7 elicited responses in wild type and cVA responses in the mutant, are missing. Without these controls the results are not interpretable. The authors should perform them, and provide an anatomical verification with immunohistochemistry.

5) Overall lack of clarity in figures. Several figures are missing essential details and notations including stimulus bars and time scales, and figure captions are missing explanations of timing. Also missing are necessary information about statistical tests including p-values. The authors also claim some results are significant without providing a statistical test in support (for example, see line 403).

6) Odd exclusion criteria for physiology data. The authors mention some of the recordings were excluded as the ORNs were not able to follow the stimulus sequence. This is an improper criterion for excluding results because it may inadvertently limit the range of results considered by the authors.

Specific questions:

- Does the manuscript have technical or conceptual flaws that should prohibit its publication? If so, please provide details.

The overly simplistic model for the moth ORNs makes it difficult to know if its conclusions are physiologically relevant.

- Are the conclusions original? If not, please provide relevant references.

The authors draw a novel conclusion from their minimalist ORN model, and show novel moth physiology data where ORNs are inhibitory on the offset of the response. However, similar observations have been observed in other species, (Kim et al 2023, Ackles et al 2021, Hallem and Carlson 2006, and several papers from the Hildebrand group).

- Do you feel that the results presented are of immediate relevance for people in your own discipline or for a broader audience? If you recommend publication, please outline briefly what you consider to be the outstanding features.

Because the authors appear to mischaracterize their own results and depend upon this mischaracterization in building their model, it is hard to know whether results are physiologically relevant. This limits the value of the manuscript for my own field or a broader audience.

Reviewer #2 (Remarks to the Author):

How ORNs encode stimulus duration is still an open question. Different observations led to different ideas, but few carefully considered the odor delivery part of the story. The authors of this manuscript took a great amount of effort to customize an olfactometer that can produce sharp odor pulses, thus minimizing or eliminating the lingering effect of odor delivery when analyzing ORN responses to odor pulses. Their data set is novel. The hardware and software they shared with the community can certainly contribute to testing further hypotheses.

Main comments

1. It's a great idea to compare the ORN responses in moths and in fruit flies because their olfactory environment is very different. Unlike moths, the fruit flies have less demand to track the fast dynamics of pheromone plume in midair. It makes total sense that the coding properties of ORNs in moths differ from those in fruit flies. This point was mentioned in the abstract and introduction, but got lost in the discussion.

2. The authors seem to suggest that the lengthened responses to short stimuli (<100ms) likely underlie the up-wind flight. While this can be a valid hypothesis, Lei et al. (<https://pubmed.ncbi.nlm.nih.gov/19232128/>) demonstrated that lengthened responses in antennal lobe produced casting-like flight behaviors in wind-tunnel.

Minor comments

1. Line 44-45: too many technical details for the introduction.

2. Line 68: It would be great to add some more explanations on the different needs between moths and flies for plume tracking.
3. Line 69-78: could be moved to Results.
4. Line 96: This sentence is a little confusing. I thought the point was that the spike generator in moth ORNs supports long range communication and navigation, but not the fly ORNs.
5. Line 214: 23 sensilla from how many animals?
6. Line 229: a frozen noise sequence?
7. Line 444: This sentence seems to suggest that *Drosophila* ORNs also have spike generating mechanisms that can produce phasic-tonic responses. True?

Reviewer #3 (Remarks to the Author):

In this manuscript Barta and colleagues investigate the encoding of stimulus duration by moth olfactory receptor neurons (ORNs) using a new odor-delivery device that delivers sharp odor pulses. The authors demonstrate that spike frequency adaptation at two different timescales allows for concentration-independent encoding of stimulus duration. They propose a new model that accurately captures the neural mechanisms involved in these physiological events. In my assessment, the manuscript holds significant interest for the scientific community, particularly those focused on chemosensory research. There are a few minor concerns that require attention.

1. In the Insects section of Material and Methods there is only information about the different species of moths used but not about *Drosophila*.
2. In Figure 2 there is a schematic illustration where the auxiliary cells are connected to the soma of the neuron through gap-junctions. Do authors mean tight junctions instead of gap-junctions? In Gu and Rospars 2011, they mention tight junctions in this position (not gap-junctions) that keep separate sensillar lymph from hemolymph.
3. In Figure 8 there is no time-scale.
4. In some Figure legends there is information missing such as the meaning of shaded areas (e.g. in Figure 8). Please, check every legend.
5. In some parts of the manuscript it is a bit confusing the way that Figures are presented and mentioned in the text (e.g. Figure 4G is mentioned after Figure 5, and Figure 7 is mentioned before Figure 6). I would recommend modifying this issue.

6. The authors show that response properties are maintained with different odor concentrations but they mostly look at the transient inhibition phase, are the spike frequencies of the rebound phase different among concentrations?

7. Is the rebound phase also happening in *S.littoralis*? It would be helpful to see the same type of raster plots in Figure 4E and F but for *S. littoralis*.

5. In this manuscript most of the conclusions arise from the work done on *A.ipsilon*, it would be extremely interesting to use this newly designed odor-delivery device in *Drosophila* to see if there are maybe more similarities between *Drosophila* and moths than expected.

Point by point response to Reviewers' comments

We thank the reviewers for carefully reading our manuscript. We improved the manuscript as per their suggestions, which we believe resulted in a more solid and easier to read work. We added three supplementary figures to provide additional information to the reader, and we moved two figures / panels from the main text to supplementary figures. The individual comments from reviewers are addressed below.

Reviewer 1

Major concern 1: Misrepresentation of the ORN responses

The authors recorded responses of moth ORNs to pulses of pheromones delivered in durations ranging from 3ms to 5s. But their figures show they inaccurately describe their own results and then base their model on these inaccuracies. For example, the repeated statement that short duration stimuli elicit prolonged responses while long stimuli do not, doesn't seem to be true. A more accurate description would be that pheromones elicit a brief burst of activity upon stimulus onset.

The authors did not justify their choices of analysis windows for inhibitory and rebound responses, and did not explain why there are gaps in between the stimulus offset and these defined windows. The authors further claim that ORN responses can be explained by two opposing forces, but do not explain the reasoning behind or significance of this claim. Overall, the representation of the data is very confusing and seemingly at odds with the results.

While the responses to brief stimuli can be classified as a burst, we disagree that our claim the responses are prolonged is not true. Regardless of the nomenclature, the ORN responses to brief stimuli typically end 80-100ms after the stimulus offset. In the new version of the manuscript, we describe the response in more detail. We say that the response to the brief stimuli can be seen as a pulse response, the shape of which is nearly identical for all brief stimuli.

We thank the reviewer for pointing out the missing information about the choice of analysis windows. We added this information in the text.

By two opposing forces, we meant the long-lasting rebound activity, which is transiently overpowered by the inhibition. We agree that this statement was confusing and was not bringing any substantial additional information, so we removed it.

Major concern 2: Lack of novelty in adaptation work

Kim et al 2023 showed that, in locusts, ORNs undergo different types of adaptation and describe how they affect responses to plume-like odor pulses. The authors should cite and describe this closely-related work.

We now mention this work in the new version of the manuscript, but it is out of scope of the present paper to review the ORN temporal patterns across insect species.

Moths are important model organisms for studies of odor-guided navigation. Our work shows that the moth olfactory system is more similar to the *Drosophila* olfactory system than previously thought. Moreover, we show a new type of response in the pheromone sensitive *Drosophila* ORNs, which is of great neuroethological interest because traditionally temporal response patterns in *Drosophila* were studied in ORNs involved in searching for food and oviposition sites, and not in pheromone sensitive neurons, associated with mate finding.

Major concern 3: Oversimplification of ORN model

One of the manuscript's main points is that ORNs can encode stimulus duration. The authors focus on the structure of the ORN and a possible mechanism that drives different types of adaptations. However, the model includes little to no detail about the structure of the sensillum or the relationships between ORNs. Similarly, the model ignores distinctions among different types of ORNs that are recorded in a sensillum. The ORNs are agnostically treated as identical units, but previous studies in various species (Hallem and Carlson 2006, Raman et al 2010, Kim et al 2023) have demonstrated the great heterogeneity in ORN responses. Because the model oversimplifies the ORN responses it is difficult to know whether its conclusions are physiologically relevant.

The vast majority of the recordings from the Z7-sensitive sensilla of *A. ipsilon* shows only a single spike amplitude [1], [2]. We agree that this is important to mention and we do so in the new version of the manuscript. Subsequently, no ephaptic coupling between ORNs housed within a sensillum occurs, because the modification of the response of one ORN requires the other ORN to be activated [3], which is highly unlikely due to the high selectivity of the pheromone-sensitive ORNs [1], [2].

While the ORNs do exhibit a certain degree of heterogeneity, this heterogeneity is well captured by our model and is indicated by the distribution of the fitted parameters in Fig. 9B. It has also been shown previously that the responses vary more in their amplitude, than in their shape [4]. To further illustrate the heterogeneity, we added a new supplementary figure (Fig. S3) with analysis of the heterogeneity of the responses.

Major concern 4: Lack of clarity in *Drosophila* data

An experiment with wild type and mutant Drosophila is shown to illustrate “phasi-tonicity” is not receptor dependent. However, the experiment is missing an essential set of negative controls: Z7 elicited responses in wild type and cVA responses in the mutant, are missing. Without these controls the results are not interpretable. The authors should perform them, and provide an anatomical verification with immunohistochemistry.

Vandroux et al. [5] already validated AipsOR3's Z7 response in modified T1 sensilla of *Drosophila* that lacks the endogenous Or67d gene. The control, i.e., mutant T1 sensilla lacking Or67d has been shown to have neither any spontaneous spiking activity, nor moth pheromone-induced evoked activity (Kurtovic et al. [6], 2007, tested with [E,Z]-10,12-hexadecadien-1-ol and (Z)-11-hexadecenal), nor *Drosophila* pheromone cVA-induced evoked activity. If deemed necessary, we can also perform the control stimulation with Z7-12:Ac, to verify that no response is evoked.

Major concern 5: Overall lack of clarity in figures

Several figures are missing essential details and notations including stimulus bars and time scales, and figure captions are missing explanations of timing. Also missing are necessary information about statistical tests including p-values. The authors also claim some results are significant without providing a statistical test in support (for example, see line 403).

Our claim on line 403 was meant to refer to Figure 4, panel F. We fixed the mentioned issues by giving a clear information about all the tests in the main text. We also included the missing information about time scales in all figures and legends.

Major concern 6: Odd exclusion criteria for physiology data

The authors mention some of the recordings were excluded as the ORNs were not able to follow the stimulus sequence. This is an improper criterion for excluding results because it may inadvertently limit the range of results considered by the authors.

Our exclusion criteria were based on the observation that some neurons changed their temporal pattern with time during the experiment. This change was observed more often with more invasive recording methods, and therefore we concluded that this change is a result of damage to the neuron. Unfortunately, it is rare in biological sciences for all samples to be usable and we tried to be as honest about our exclusion criteria as possible.

Reviewer 2

Reviewer 2 wrote two main points, which we discuss below, together with minor comments 4 and 7, which are related to the first main point.

Main point 1 + Minor comments 4 and 7:

It's a great idea to compare the ORN responses in moths and in fruit flies because their olfactory environment is very different. Unlike moths, the fruit flies have less demand to track the fast dynamics of pheromone plume in midair. It makes total sense that the coding properties of ORNs in moths differ from those in fruit flies. This point was mentioned in the abstract and introduction, but got lost in the discussion.

Line 96: This sentence is a little confusing. I thought the point was that the spike generator in moth ORNs supports long range communication and navigation, but not the fly ORNs.

*Line 444: This sentence seems to suggest that *Drosophila* ORNs also have spike generating mechanisms that can produce phasic-tonic responses. True?*

Some *Drosophila* ORNs also have spike generating mechanism that can produce phasi-tonic response, typically ORNs involved in searching for food and oviposition sites. We refer to the phasi-tonic response in *Drosophila* ORNs also on lines 41-42 and 609-610. It has been shown in *Drosophila* that this phasi-tonicity originates in the spike generating mechanism [7]. We find this connection very important, because due to the vast genetic tools available for *Drosophila* it is likely that the exact molecular mechanisms of the spike generator will be uncovered first in *Drosophila* rather than in moths. Although some differences remain, the similarity indicates that there is also a similarity in the molecular mechanisms.

On the other hand, we observe a striking difference of the responses of pheromone-sensitive ORNs between moths and *Drosophila*. Unlike ORNs that exhibit a phasi-tonic response, *Drosophila* pheromone sensitive ORNs are not involved in long range navigation, only for recognition on very short distances. There is, therefore, a striking difference in behavioral significance of these ORNs. In our manuscript, we show that this difference in behavioral significance translates to difference in temporal response patterns.

In the new version of the manuscript, we now discuss this more in the discussion to highlight this important finding.

Main point 2

The authors seem to suggest that the lengthened responses to short stimuli (<100ms) likely underlie the up-wind flight. While this can be a valid hypothesis, Lei et al. (<https://pubmed.ncbi.nlm.nih.gov/19232128/>) demonstrated that lengthened responses in antennal lobe produced casting-like flight behaviors in wind-tunnel.

We thank the Reviewer for pointing out this relevant study to us. We now discuss this study in the manuscript.

Minor comment 1

Line 44-45: too many technical details for the introduction.

Removed.

Minor comment 2

Line 68: It would be great to add some more explanations on the different needs between moths and flies for plume tracking.

Added.

Minor comment 3

Line 69-78: could be moved to Results.

We thank the Reviewer for the suggestion. While we agree that we are already including some results in the Introduction section, we believe that by keeping this paragraph in the Introduction section, the Introduction section reads better. We can modify this, if Reviewers disagree and think that it should be moved.

Minor comment 5

Line 214: 23 sensilla from how many animals?

In all experiments, we recorded from 1-3 sensilla from each animal. We now included this information in the manuscript.

Minor comment 6

A frozen noise sequence?

By frozen noise sequence, we mean that a 2-second-long stimulus sequence was generated, and this stimulus sequence was presented to the animal multiple times. We removed the word *frozen* and tried to be clearer about the used stimulus sequence.

Reviewer 3

The comments 1, 3, 4, and 5 regard the legibility of the figures and the manuscript. We are thankful to the Reviewer for pointing out these issues. In the following we discuss the remaining comments.

Comment 1

In the Insects section of Material and Methods there is only information about the different species of moths used but not about Drosophila.

Corrected.

Comment 2

In Figure 2 there is a schematic illustration where the auxiliary cells are connected to the soma of the neuron through gap-junctions. Do authors mean tight junctions instead of gap-junctions? In Gu and Rospars 2011, they mention tight junctions in this position (not gap-junctions) that keep separate sensillar lymph from hemolymph.

Yes, that is correct. It should be “tight junctions”. We corrected it in the figure.

Comment 3

In Figure 8 there is no time-scale.

Corrected.

Comment 4

In some Figure legends there is information missing such as the meaning of shaded areas (e.g. in Figure 8). Please, check every legend.

The shaded area represents a confidence band obtained by bootstrapping the responses. We now describe this in the legend. We checked the legends and filled in the missing information.

Comment 5

In some parts of the manuscript it is a bit confusing the way that Figures are presented and mentioned in the text (e.g. Figure 4G is mentioned after Figure 5, and Figure 7 is mentioned before Figure 6). I would recommend modifying this issue.

The shaded area represents a confidence band obtained by bootstrapping the responses. We now describe this in the legend. We moved both Fig. 4G and Fig. 7 to supplementary figures.

Comment 6

The authors show that response properties are maintained with different odor concentrations but they mostly look at the transient inhibition phase, are the spike frequencies of the rebound phase different among concentrations?

Yes, the spike frequency during rebound activity increases with concentration. This is visible in Fig. 5B, in green. We now report in the text the Spearman correlation coefficient for the three

different durations to verify that the contrast between the rebound and inhibitory activity grows with stimulus dose.

Comment 7

*Is the rebound phase also happening in *S. littoralis*? It would be helpful to see the same type of raster plots in Figure 4E and F but for *S. littoralis*.*

Yes, the rebound phase also occurs in *S. littoralis*. We modified the figure in the manuscript accordingly, as requested by the reviewer.

Comment 8

*In this manuscript most of the conclusions arise from the work done on *A. ipsilon*, it would be extremely interesting to use this newly designed odor-delivery device in *Drosophila* to see if there are maybe more similarities between *Drosophila* and moths than expected.*

In our work, we showed that the moth ORN responses are more similar to *Drosophila* ORN responses than previously thought. Most of the odor molecules typically used to study the response dynamics of *Drosophila* ORNs have very high volatility and the distortion of odor delivery dynamics is therefore not as important as with the pheromone compounds we used. We tested the response dynamics of *Drosophila* ORNs to the pheromone cVA, which has a very low volatility, and therefore typically suffers of the same issues as moth pheromone molecules. Surprisingly, we found that cVA sensitive ORNs do not have the same firing response pattern as other *Drosophila* ORNs. We agree that showing the similarity between *Drosophila* and other model organisms is important, due to the availability of vast genetic tools in *Drosophila*. However, we think that subsequently showing the differences is just as important, as it can provide clues about how the different responses adapted for different environments and needs of the animal.

- [1] M. Renou, C. Gadenne, and D. Tauban, "Electrophysiological investigations of pheromone-sensitive sensilla in the hybrids between two moth species," *J. Insect Physiol.*, vol. 42, no. 3, pp. 267–277, Mar. 1996, doi: 10.1016/0022-1910(95)00108-5.
- [2] D. Jarriault, C. Gadenne, P. Lucas, J.-P. Rospars, and S. Anton, "Transformation of the Sex Pheromone Signal in the Noctuid Moth *Agrotis ipsilon*: From Peripheral Input to Antennal Lobe Output," *Chem. Senses*, vol. 35, no. 8, pp. 705–715, Oct. 2010, doi: 10.1093/chemse/bjq069.
- [3] C.-Y. Su, K. Menuz, J. Reisert, and J. R. Carlson, "Non-synaptic inhibition between grouped neurons in an olfactory circuit," *Nature*, vol. 492, no. 7427, pp. 66–71, Dec. 2012, doi: 10.1038/nature11712.
- [4] J.-P. Rospars *et al.*, "Heterogeneity and Convergence of Olfactory First-Order Neurons Account for the High Speed and Sensitivity of Second-Order Neurons," *PLoS Comput. Biol.*, vol. 10, no. 12, p. e1003975, Dec. 2014, doi: 10.1371/journal.pcbi.1003975.

- [5] P. Vandroux *et al.*, "Activation of pheromone-sensitive olfactory neurons by plant volatiles in the moth *Agrotis ipsilon* does not occur at the level of the pheromone receptor protein," *Front. Ecol. Evol.*, vol. 10, Nov. 2022, doi: 10.3389/fevo.2022.1035252.
- [6] A. Kurtovic, A. Widmer, and B. J. Dickson, "A single class of olfactory neurons mediates behavioural responses to a *Drosophila* sex pheromone," *Nature*, vol. 446, no. 7135, pp. 542–546, Mar. 2007, doi: 10.1038/nature05672.
- [7] K. I. Nagel and R. I. Wilson, "Biophysical mechanisms underlying olfactory receptor neuron dynamics," *Nat. Neurosci.*, vol. 14, no. 2, pp. 208–216, Feb. 2011, doi: 10.1038/nn.2725.

Reviewers' comments:

Reviewer #1 (Remarks to the Author):

With this revision the authors have satisfied most of my main concerns by adding new data and revising the text. But I still have two substantial concerns:

(1) The authors still need to perform a negative control with Z7-12:Ac. In their response to my major concern #4 the authors wrote "if deemed necessary, we will perform the control stimulation with Z7-12:Ac, to verify that no response is evoked." This control is necessary and should be performed.

(2) I remain concerned about the way the authors chose to exclude some data from their analysis (my major concern #6). The authors explained that they excluded some results after collecting them because the results were unexpected (e.g. response patterns changed over time). This is an improper exclusion criterion because this procedure will necessarily reinforce the authors' biases. The situation described by the authors is very common. Electrophysiologists deal with it this way: (1) don't begin the experiment trial until the recording is stable; (2) once the trial begins, if the results include variability, collect more data. To remedy this, the authors should collect more data and include all of it in the analysis.

Reviewer #2 (Remarks to the Author):

My questions are all answered, but I still have some minor points to add.

Line 275, I believe it is 3 spikes/sec.

Line 635: A main discovery of Lei et al. (2009) was that a disruption of the inhibitory period after each pulsatile excitation resulted in disruption of plume-tracking behavior in the moth. This is consistent with the main results of this manuscript. In the moth data, whether the stimulation is short or long, the response pattern of ORNs is consisted of excitatory period followed by inhibitory period. In my opinion, both periods are important for the plume-tracking behavior, but it is undetermined which component of the tracking behavior is correlated with what phase of the responses. Regardless it is at the ORN level or antennal lobe level, these responses are still at the early processing stage of the olfactory pathway.

Reviewer #3 (Remarks to the Author):

In my opinion, the authors have revised the manuscript to a standard that satisfies concerns raised and it is ready for publication.

Minor points:

Line 57 – although VPC accounts for volatile plant compound, it is not defined anywhere in the text.

Line 642- the bibliographic reference is missing.

Point by point response

Reviewer 1

Point 1: *The authors still need to perform a negative control with Z7-12:Ac. In their response to my major concern #4 the authors wrote "if deemed necessary, we will perform the control stimulation with Z7-12:Ac, to verify that no response is evoked." This control is necessary and should be performed.*

We performed the control experiment and added it as a supplementary figure.

Point 2: *I remain concerned about the way the authors chose to exclude some data from their analysis (my major concern #6). The authors explained that they excluded some results after collecting them because the results were unexpected (e.g. response patterns changed over time). This is an improper exclusion criterion because this procedure will necessarily reinforce the authors' biases. The situation described by the authors is very common. Electrophysiologists deal with it this way: (1) don't begin the experiment trial until the recording is stable; (2) once the trial begins, if the results include variability, collect more data. To remedy this, the authors should collect more data and include all of it in the analysis.*

We did not exclude recordings because they were unexpected but because they were unstable. One of the risks of single sensillum recordings with glass electrodes is that a leak of ringer from the recording electrode modifies the activity of ORNs. This fits well with the observation that recordings were extremely stable with tungsten electrode, stable most of the time with glass electrode inserted at the base of sensilla and unstable with glass electrodes in contact with the cut tip of sensilla, which is why we didn't use tip recordings in this work. In all recordings the response shape was the same at the beginning, but in some cases it changed over time. All stable tungsten and base recordings showed the same response shape. Therefore, we disagree that we used improper exclusion criterion of unstable recordings, which represent a minority of the recordings, and that it biased our dataset.

Reviewer 2

Point 1: *Line 275, I believe it is 3 spikes/sec.*

It is 3 spikes / ms. We assume that each PN is receiving input from a large number of ORN which combine to an input of 3 spikes / ms.

Point 2: *Line 635: A main discovery of Lei et al. (2009) was that a disruption of the inhibitory period after each pulsatile excitation resulted in disruption of plume-tracking behavior in the moth. This is consistent with the main results of this manuscript. In the moth data, whether the stimulation is short or long, the response pattern of ORNs is consisted of excitatory period*

followed by inhibitory period. In my opinion, both periods are important for the plume-tracking behavior, but it is undetermined which component of the tracking behavior is correlated with what phase of the responses. Regardless it is at the ORN level or antennal lobe level, these responses are still at the early processing stage of the olfactory pathway.

We agree that we made an assumption and corrected it in line 620 (manuscript with highlighted changes): “It is not yet determined which component of the plume-tracking behavior is correlated with what phase of the responses but we hypothesize that inhibitory periods are necessary for the transition from upwind flight to casting behavior.”

Reviewer 3

Point 1: *Line 57 – although VPC accounts for volatile plant compound, it is not defined anywhere in the text.*

Thank you for pointing it out. Corrected.

Point 2: *Line 642- the bibliographic reference is missing.*

This point is unclear to us. Belmabrouk et al. is referenced on the line 592 (manuscript with highlighted changes).

REVIEWERS' COMMENTS:

Reviewer #1 (Remarks to the Author):

I thank the authors for performing an additional control experiment, and for explaining their criteria for excluding data from their analysis. I have no further concerns about this manuscript.